# Navigating the Accuracy-Size Trade-Off with Flexible Model Merging

**Akash Dhasade**[1],[*] **Divyansh Jhunjhunwala**[2]**, Milos Vujasinovic**[1]**, Gauri Joshi**[2]**,**
**Anne-Marie Kermarrec**[1]
[1]EPFL, Switzerland   [2]Carnegie Mellon University, USA
akash.dhasade@epfl.ch   milos.vujasinovic@epfl.ch

## Abstract

Model merging has emerged as an efficient method to combine multiple single-task fine-tuned models. The merged model can enjoy multi-task capabilities without expensive training. While promising, merging into a single model often suffers from an accuracy gap with respect to individual fine-tuned models. On the other hand, deploying all individual fine-tuned models incurs high storage costs. We propose FLEXMERGE, a novel data-free model merging framework that: *(a)* flexibly generates merged models of varying sizes, spanning the full spectrum from a single merged model to retaining all individual fine-tuned models; and *(b)* supports multiple merging algorithms in a unified framework. Using FLEXMERGE, we systematically characterize the accuracy–size trade-off of different algorithms. Our study reveals two key findings: first, even modestly larger merged models can yield steep accuracy gains (up to $13.5\%$ when just doubling the size); second, algorithm rankings are not consistent as size increases, with some methods overtaking others beyond the one-model regime. These results uncover a new design dimension for model merging: developing and comparing algorithms across the full spectrum of sizes rather than only at the single-model limit. Extensive experiments on vision and NLP benchmarks, with up to 30 tasks, confirm the generality and practicality of FLEXMERGE.[1]

## 1 Introduction

In recent years, the pre-training followed by fine-tuning paradigm has become the leading approach in both natural language processing (NLP) and computer vision, showcasing remarkable success on a wide range of tasks (Devlin et al., 2018; Dodge et al., 2020; Dosovitskiy et al., 2021; Bommasani et al., 2021). Pre-trained models (PTMs), which learn generalized features from large-scale datasets, serve as powerful starting points, enabling fine-tuning to achieve superior performance on downstream tasks with less labeled data. This has led to an exponential growth in the number of fine-tuned models driven further by the availability of open-source repositories (maintainers & contributors, 2016; Wolf et al., 2019). However, *deploying individual fine-tuned models* for specific tasks incurs high storage and deployment costs. The alternative is Multi-task learning (MTL), which aims to jointly train *a single model* across multiple tasks (Vandenhende et al., 2021; Sanh et al., 2022). But MTL comes with its own drawbacks, such as significant computational overhead and the need to simultaneously access the data from all tasks, which might be infeasible due to privacy constraints (Jin et al., 2023).

To mitigate these limitations, model merging has emerged as a promising solution, allowing the combination of multiple fine-tuned models into a *single model* without access to training data. To this end, several model merging methods have been proposed (Gargiulo et al., 2025; Huang et al., 2024; Yang et al., 2024a; Yadav et al., 2023; Ilharco et al., 2023; Matena & Raffel, 2022). However, a single model is often unable to perfectly resolve parameter conflicts between tasks, leaving an accuracy gap with respect to the individual fine-tuned models (Zhang et al., 2025; Huang et al., 2024). This gap becomes more significant as a higher number of models are merged (Yadav et al., 2023; Ilharco et al., 2023). To mitigate this issue, some methods leverage additional data to facilitate merging (Lu et al.,

---

[*]Work done during research visit to Carnegie Mellon University.
[1]Our code is available at `https://github.com/sacs-epfl/flexmerge`

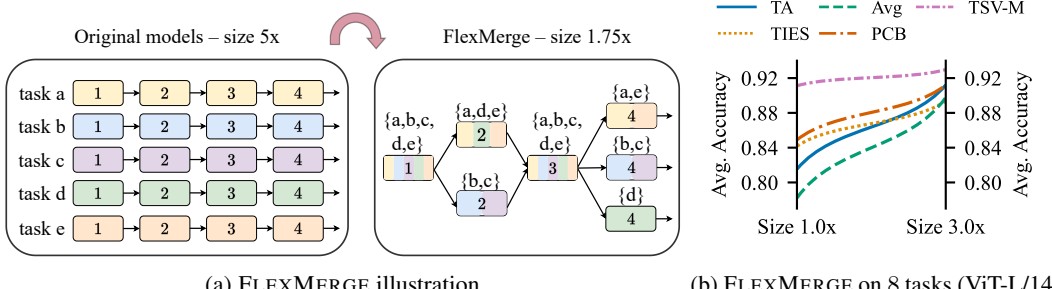

(a) FLEXMERGE illustration

(b) FLEXMERGE on 8 tasks (ViT-L/14)

Figure 1: (a) Fine-tuned models are sequences of blocks. FLEXMERGE iteratively merges block pairs until reaching the desired size (*e.g.*, size $1.75\times$). (b) Algorithm rankings change as size is increased.

2024; Yang et al., 2024a; Tang et al., 2024a; Yang et al., 2024b). Yet, the data-dependency might be difficult to meet in practice due to privacy constraints or proprietary restrictions, leading to a growing focus on data-free model merging techniques (Gargiulo et al., 2025; Huang et al., 2024; Du et al., 2024; Yu et al., 2024; Yadav et al., 2023). Nevertheless, in the absence of data, the accuracy gap remains significant, highlighting the need for novel solutions.

We argue that an effective solution to this challenge is to go beyond the conventional one model approach, and merge into model(s) of bigger sizes. Merging multiple fine-tuned models naturally presents a trade-off between maintaining accuracy and achieving model compactness, dictated by the size of the merged model. This trade-off spans a spectrum: at one extreme, retaining all individual fine-tuned models for each task achieves maximal accuracy but at the cost of larger overall size; at the other, fully merging all tasks into a single model minimizes storage size but sacrifices accuracy. Despite this clear trade-off, a systematic investigation of the accuracy-size relationship in model merging has been lacking. In this light, we pose two key research questions: *(RQ1) How can we derive merged models across the full range of model sizes in a data-free manner?* and *(RQ2) What is the nature of the accuracy-size trade-off exhibited by different data-free merging algorithms?*

In response to (RQ1), we propose FLEXMERGE, a flexible framework that enables *data-free* fusion into model(s) of any desired size. At its core, FLEXMERGE treats each fine-tuned model as composed of sequential blocks, as illustrated in Figure 1(a), whose granularity can be controlled (*e.g.*, a transformer block, a few layers, or even a single layer). It then takes a bottom-up approach starting with all fine-tuned models with their respective blocks and greedily merging a pair of blocks with the highest cosine similarity in each merging iteration. This merging can leverage *any* existing data-free merging method such as Task Arithmetic (TA) (Ilharco et al., 2023), TIES-MERGING (Yadav et al., 2023), EMR-MERGING (Huang et al., 2024), TSV-M (Gargiulo et al., 2025), *etc.*, applied at the block-level. With each merging iteration, the size of the deployed model is reduced, and the process can be halted once the desired size is met. For instance, in Figure 1(a), the merging is halted when the merged model is $1.75\times$ the size of a single fine-tuned model. The entire merging process in FLEXMERGE needs no additional data or tuning, making FLEXMERGE fully *data-free*.

In response to (RQ2), we demonstrate with FLEXMERGE that a range of data-free merging algorithms exhibit highly favorable accuracy-size trade-offs. Remarkably, the accuracy-size trade-off is characterized by steep gains in accuracy for even modestly bigger merged models beyond one model, followed by steady improvements, reaching near fine-tuning accuracy well before the maximum size. To illustrate this in practice, Figure 2 charts the merged model accuracy versus deployed size for 8 tasks (top) and 30 tasks (bottom) using the ViT-B/32 model, with TA (Ilharco et al., 2023) and

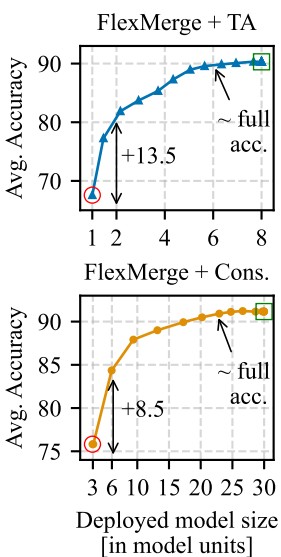

Figure 2: FLEXMERGE enables large accuracy gains when just doubling the deployed model size and attains full accuracy well before the maximum size.

CONSENSUS (Wang et al., 2024) as the respective merging methods. ○ and □ annotate the accuracy at both ends of the spectrum *i.e.*, lowest fused size and retaining all fine-tuned models respectively. FLEXMERGE + TA gains $13.5\%$ in average accuracy when going from $1\times$ to $2\times$ while FLEXMERGE + CONSENSUS gains $8.5\%$ when doubling the size from approximately $3\times$ to $6\times$. We note that CONSENSUS requires storing masks and the pre-trained parameters alongside the unified parameters (Wang et al., 2024), resulting in the lowest possible size of $\approx 3\times$ for 30 tasks. We observe that the steep rise is followed by relatively slower accuracy growth in the middle. Yet, a near fine-tuning accuracy is attained well before the maximum size. For 8 tasks, this is obtained around size $6\times$ and for 30 tasks, around size $23.5\times$. Secondly, we observe that algorithm rankings are not consistent even at modestly bigger sizes. Figure 1(b) shows that vanilla averaging exceeds TIES-MERGING while TA attains the performance of PCB-MERGING at size $3\times$ despite starting from a large gap at $1\times$. *Our findings open a new design dimension: encouraging algorithm development and comparison for sizes $> 1\times$ instead of restricting only to $1\times$.*

**Contributions.** To the best of our knowledge, we present the first study of model merging that:

- Generates merged models across full spectrum of sizes, *including non-integer sizes*;
- Supports a wide range of data-free merging algorithms, *within a unified framework*;
- Provides a systematic characterization of the accuracy-size trade-off in data-free model merging, *revealing general trends, highly favorable regions and inconsistency of algorithm rankings*;
- Demonstrates that larger merged sizes incur negligible inference-time overhead, *enabled by our efficient implementation*.

We confirm our findings through extensive experiments spanning language and vision modalities, multiple model families, multi-modal datasets, using both full-parameter fine-tuning (FFT) and parameter efficient fine-tuning (PEFT), scaling up to 30 tasks.

## 2 RELATED WORK

Initial studies on model merging focused on vanilla averaging as a way of combining models obtained from same or different training runs of a task into one higher performing model (Izmailov et al., 2018; Gupta et al., 2020; Wortsman et al., 2022; Cha et al., 2021). Vanilla averaging is also used in federated learning to merge different client models (McMahan et al., 2017; Konečný et al., 2016). Ilharco et al. (2023) introduced task vectors, representing the difference between fine-tuned and pre-trained models, enabling model combination through vector arithmetic.

**Data-based merging** methods leverage validation data to facilitate merging. Techniques like FISHER MERGING (Matena & Raffel, 2022) and REGMEAN (Jin et al., 2023) compute the Fisher Information and Gram matrices, respectively, for weighted averaging of model parameters. SURGERY (Yang et al., 2024a) trains task-specific adapters to debias the representations produced by the merged model. ADAMERGING (Yang et al., 2024b) introduces per-task, per-layer merging co-efficients, and proposes to learn these co-efficients by solving an entropy minimization objective. WEMOE (Tang et al., 2024a) merges all modules except for task-specific MLPs, which are retained as weight-ensembled mixture-of-experts (MoE) with learned routers. TWIN-MERGING (Lu et al., 2024) leverages MoE on difference vectors *i.e.*, the difference between the fine-tuned models and the merged model. While the availability of validation data enhances accuracy, such data might be difficult to obtain in practice.

**Data-free merging** directly merges model parameters without any data. TIES-MERGING (Yadav et al., 2023) resolves parameter interference by trimming redundant parameters and resolving sign conflicts. PCB-MERGING (Du et al., 2024) considers both intra- and inter-parameter competition balancing. DARE (Yu et al., 2024) reduces parameter interference by randomly dropping parameters and proportionally rescaling remaining ones. EMR-MERGING (Huang et al., 2024) introduces the paradigm of maintaining light-weight task specific masks in addition to the merged model to enhance performance. CONSENSUS (Wang et al., 2024) also relies on task specific masks, but creates them differently compared to EMR-MERGING. Both approaches significantly improve accuracy over previous methods, albeit at the cost of test-time reconstruction overhead (Gargiulo et al., 2025). TSV-M (Gargiulo et al., 2025) merges SVD-decomposed task singular vectors, reducing interference by retaining only prominent singular directions and orthogonalizing them across tasks.

Recent work by Zhang et al. (2025) explores merging into sizes $> 1\times$. Their method, CHANNEL MERGING, relies on layer-wise K-Means clustering followed by merging within each cluster using only TA. However, this approach is restrictive as it cannot generate fractional-sized models. Despite the emergence of advanced methods and attempts at merging into bigger sizes, to the best of our knowledge, no prior work has systematically investigated the accuracy–size trade-off in model merging under a single unified framework. For completeness, we provide additional related work and a taxonomy of existing algorithms based on their data-free/data-based nature in Appendix A.

## 3 FLEXMERGE

### 3.1 PRELIMINARIES

We consider a set of $M$ tasks: $\{T_1, \ldots, T_M\}$, where the fine-tuned model parameters for task $T_i$ are denoted by $\boldsymbol{\theta}_i$. These fine-tuned parameters are typically obtained by adapting a pre-trained model, such as ViT (Dosovitskiy et al., 2021) or T5 (Raffel et al., 2020) using either full parameter fine-tuning (FT) or parameter-efficient fine-tuning (PEFT) methods (Liu et al., 2022). Thus, it is assumed that all the fine-tuned models have the same size and the model architecture as the pre-trained model, as also considered in prior work (Ilharco et al., 2023; Yadav et al., 2023). To analyze the changes introduced by fine-tuning, we use the concept of task vectors $\boldsymbol{\tau}_i$ introduced by Ilharco et al. (2023), where $\boldsymbol{\tau}_i = \boldsymbol{\theta}_i - \boldsymbol{\theta}_{\text{pre}}$, with $\boldsymbol{\theta}_{\text{pre}}$ being the pre-trained weights. These task vectors capture the specific modifications needed for each task and provide a compact representation for merging.

Standard model merging approaches involve combining the task-vectors $\{\boldsymbol{\tau}_1, \ldots, \boldsymbol{\tau}_M\}$ into a unified task vector $\boldsymbol{\tau}_{\text{uni}} = \mathcal{F}(\{\boldsymbol{\tau}_1, \ldots, \boldsymbol{\tau}_M\})$ and then adding the unified task vector to the pre-trained weights to get the final merged model, $\boldsymbol{\theta}_{\text{uni}} = \boldsymbol{\theta}_{\text{pre}} + \boldsymbol{\tau}_{\text{uni}}$. Here $\mathcal{F}$ denotes the merging algorithm used to obtain the unified task vector's weights. For example, the unified task vector $\boldsymbol{\tau}_{\text{uni}}$ can be computed via simple averaging $\boldsymbol{\tau}_{\text{uni}} = \frac{1}{M} \sum_{i=1}^{M} \boldsymbol{\tau}_i$ or via TA (Ilharco et al., 2023) that uses a coefficient $\lambda$ to weigh the contribution[2] of the unified task vector $\boldsymbol{\tau}_{\text{uni}} = \lambda \cdot \frac{1}{M} \sum_{i=1}^{M} \boldsymbol{\tau}_i$ in the final merged model. It is shown that just by tuning $\lambda$, one can outperform weight averaging (Ilharco et al., 2023).

**Motivation.** Merging into one model $\boldsymbol{\theta}_{\text{uni}}$ may cause accuracy deterioration due to parameter interference between different fine-tuned models (Zhang et al., 2025; Yadav et al., 2023). This behavior becomes prominent as more and more fine-tuned models are merged, as discussed in Section 1. On the other hand, retaining all fine-tuned models preserves full fine-tuning accuracy but results in a net size $M\times$ that of one fine-tuned model, which is impractical due to the high memory requirements. In this work, we investigate the problem of generating models of any desired size in the range $[1, M]$, including models with fractional size such as $2.25\times$ model units.

### 3.2 PROPOSED APPROACH

To enable a more granular fusion, we consider the model to be composed of $B$ sequential blocks, for instance transformer blocks in a ViT model or even layers within each transformer block such as attention or MLP layers could be considered as unique blocks. Assuming $B$ total blocks, we consider the task vectors for each block as $\{\boldsymbol{\tau}_k^b\}_{b=1}^{B}$ corresponding to the original task vector $\boldsymbol{\tau}_k$ for a task $k$. Our proposed framework, FLEXMERGE, takes a greedy approach to efficiently merge task vectors from multiple tasks at the granularity of blocks, aiming to reduce the deployed model size while maintaining utility. The pseudo-code for FLEXMERGE is presented in Algorithm 1.

**Initialization (Lines 1–6).** The merging proceeds bottom-up. Initially, no merging has occurred, and we retain $\boldsymbol{\tau}_k^b$ for all tasks $k \in [M]$ and all blocks $b \in [B]$ (see Figure 1(a)). For each block $b$, we initialize a set of tuples: $\mathcal{G}^b = \{(\{k\}, \boldsymbol{\tau}_k^b) \mid k \in [M]\}$. Each tuple in $\mathcal{G}^b$ consists of: *(i)* a task set $\{k\}$ (tracking which tasks are represented) and *(ii)* the corresponding block task vector $\boldsymbol{\tau}_k^b$. For example, in Figure 1(a) for the first block, we would have $\mathcal{G}^1 = \{(\{a\}, \boldsymbol{\tau}_a^1), \ldots, (\{e\}, \boldsymbol{\tau}_e^1)\}$. When the merging terminates, the resulting $\mathcal{G}^1$ for Figure 1(a) would be $\mathcal{G}^1 = \{(\{a, \ldots, e\}, \hat{\boldsymbol{\tau}}_{\text{uni}}^1)\}$, where $\hat{\boldsymbol{\tau}}_{\text{uni}}^1$ is the merged task vector for the first block for all tasks. The initial size $S$ is calculated as the cumulative size of all block parameters across $M$ tasks.

---

[2]We add a scaling factor of $1/M$ to the standard definition $\boldsymbol{\tau}_{\text{uni}} = \lambda \cdot \sum_{i=1}^{M} \boldsymbol{\tau}_i$ given in (Ilharco et al., 2023) to better suit its usage in FLEXMERGE where $M$ can vary across blocks.

---

**Algorithm 1:** The FLEXMERGE framework

---

**Input:** Task vectors $\{\boldsymbol{\tau}_k^b\}$ for all $k \in [M], b \in [B]$; merging algorithm $\mathcal{F}$; target size $S_{\text{target}}$
**Output:** Merged task vectors with reduced size

1   $S \leftarrow 0$          ▷ *Initialize deployed size*
2   **for** $b = 1$ **to** $B$ **do**
3     $\mathcal{G}^b \leftarrow \emptyset$
4     **for** $k = 1$ **to** $M$ **do**
5       $\mathcal{G}^b \leftarrow \mathcal{G}^b \cup \left(\{k\}, \boldsymbol{\tau}_k^b\right)$
6       $S \leftarrow S + \text{size}(\boldsymbol{\tau}_k^b)$

7   **while** $S > S_{\text{target}}$ **or** *not all blocks merged* **do**
8     Find block $b^*$ and pair $(g_{i*}, g_{j*}) \in \mathcal{G}^{b^*}$ with the highest similarity:
9

$$(b^*, g_{i*}, g_{j*}) = \underset{b \in [B],\, g_i, g_j \in \mathcal{G}^b}{\arg\max} \text{SIMILARITY}(g_i, g_j)$$

10     $\mathcal{T}_{i*}^{b^*}, \mathcal{T}_{j*}^{b^*} \leftarrow g_{i*}(0),\ g_{j*}(0)$          ▷ *Get task subsets*
11     $\mathcal{T}_{\text{uni}}^{b^*} \leftarrow \mathcal{T}_{i*}^{b^*} \cup \mathcal{T}_{j*}^{b^*}$          ▷ *Merge task subsets*
12     $\boldsymbol{\tau}_{\text{uni}}^{b^*} \leftarrow \mathcal{F}(\{\boldsymbol{\tau}_k^{b^*} \mid k \in \mathcal{T}_{\text{uni}}^{b^*}\})$          ▷ *Merge task vectors*
13     $\mathcal{G}^{b^*} \leftarrow \mathcal{G}^{b^*} \cup \left(\mathcal{T}_{\text{uni}}^{b^*}, \boldsymbol{\tau}_{\text{uni}}^{b^*}\right) \setminus \{g_{i*}, g_{j*}\}$      ▷ *Update the block*
14     $S \leftarrow S - \text{size}(\boldsymbol{\tau}_{\text{uni}}^{b^*})$          ▷ *Update current size*

---

**Iteration (lines 7-14).** In each iteration, the algorithm identifies a block $b^*$ and pair of tuples $(g_{i*}, g_{j*}) \in \mathcal{G}^{b^*}$, which have the highest similarity (as defined below). Then they are merged as follows. Let $\mathcal{T}_{i*}^{b^*}$ and $\mathcal{T}_{j*}^{b^*}$ be the subset of tasks associated with $g_{i*}$ and $g_{j*}$ respectively, *i.e.*, the first elements of $g_{i*}$ and $g_{j*}$ respectively. First, $\mathcal{T}_{i*}^{b^*}$ and $\mathcal{T}_{j*}^{b^*}$ are merged via a union operation: $\mathcal{T}_{\text{uni}}^{b^*} = \mathcal{T}_{i*}^{b^*} \cup \mathcal{T}_{j*}^{b^*}$. Next, the merged task vector corresponding to block $b^*$ and set $\mathcal{T}_{\text{uni}}^{b^*}$ is created as follows: $\boldsymbol{\tau}_{\text{uni}}^{b^*} = \mathcal{F}(\{\boldsymbol{\tau}_k^{b^*} \mid k \in \mathcal{T}_{\text{uni}}^{b^*}\})$. Here $\mathcal{F}$ can be *any* data-free merging algorithm. The tuple set $\mathcal{G}^{b^*}$ is then updated by removing the tuples $g_{i*}, g_{j*}$ and adding the new merged tuple $(\mathcal{T}_{\text{uni}}^{b^*}, \boldsymbol{\tau}_{\text{uni}}^{b^*})$. Each merge reduces the model size by the size of the task vector corresponding to block $b^*$, and the process continues until the current size $S$ meets the desired size $S_{\text{target}}$ or no further merges are possible.

**Similarity function.** We measure the similarity between two groups $g_i, g_j$ in any block $b$ using the lowest cosine similarity between any pair of original task vectors corresponding to the tasks in the sets $\mathcal{T}_i^b$ and $\mathcal{T}_j^b$:

$$\text{SIMILARITY}(g_i, g_j) = \min_{k_1 \in \mathcal{T}_i^b,\, k_2 \in \mathcal{T}_j^b} \text{cosine\_sim}(\boldsymbol{\tau}_{k_1}^b, \boldsymbol{\tau}_{k_2}^b). \tag{1}$$

Our choice of the $\min$ similarity derives from our ablations comparing different strategies—$\max$, $\min$, and average—as well as computing similarity between merged group task vectors directly. Among these, $\min$ yields the best performance. Thus at each iteration, we merge the pair of groups with the highest of these minimum similarities (line 9, Algorithm 1). While the cosine similarity between full task vectors can be relatively low (Ilharco et al., 2023), the block-level similarities tend to be higher and effective for merging. CHANNEL MERGING (Zhang et al., 2025) also employs cosine similarity.

**Enhancing efficiency.** The pairwise similarities can be precomputed once for all pairs and accessed in constant time during the merging process. Furthermore, we implement $\mathcal{G}^b$ using the Disjoint Set Union (DSU) (Cormen et al., 2009) data structure to efficiently track and unify task sets for each block. Our design enables FLEXMERGE to perform very efficient merging even under many tasks (see Table 2).

**Complexity Analysis.** Algorithm 1 identifies most similar block pairs in each iteration. We presented the algorithm in this form for conceptual clarity. However, in practice, we can generate the global merging order for all blocks first and then apply merges. To analyze the time complexity of FLEXMERGE, we consider three distinct stages of this process: similarity pre-computation, generating merging order, and actual parameter merging.

Table 1: Summary of existing data-free merging methods. Column $\mathcal{F}(\{\boldsymbol{\tau}_k^b \mid k \in \mathcal{T}_{\mathbf{uni}}^b\})$ denotes the result of merging. Figure 7 (Section B) provides an illustrative diagram.

| Algorithm | $\mathcal{F}(\{\boldsymbol{\tau}_k^b \mid k \in \mathcal{T}_{\mathbf{uni}}^b\})$ | Final Model | What is stored? |
|---|---|---|---|
| TA (Ilharco et al., 2023), TIES (Yadav et al., 2023), Avg. (Ilharco et al., 2023), PCB (Du et al., 2024), TSV-M (Gargiulo et al., 2025) | $\boldsymbol{\tau}_{\text{uni}}^b$ | $\boldsymbol{\theta}_{\text{uni}}^b = \boldsymbol{\theta}_{\text{pre}}^b + \boldsymbol{\tau}_{\text{uni}}^b$ | $\boldsymbol{\theta}_{\text{uni}}^b$ |
| CONSENSUS (Wang et al., 2024) | $\boldsymbol{\tau}_{\text{uni}}^b, \{\boldsymbol{m}_k^b \mid k \in \mathcal{T}_{\text{uni}}\}$ | $\hat{\boldsymbol{\theta}}_k^b = \boldsymbol{\theta}_{\text{pre}}^b + \boldsymbol{\tau}_{\text{uni}}^b \circ \boldsymbol{m}_k^b$ (reconstructed per-task $k$) | $\boldsymbol{\theta}_{\text{pre}}^b, \boldsymbol{\tau}_{\text{uni}}^b,$ $\{\boldsymbol{m}_k^b \mid k \in \mathcal{T}_{\text{uni}}\}$ |
| EMR-MERGING (Huang et al., 2024) | $\boldsymbol{\tau}_{\text{uni}}^b, \{\boldsymbol{m}_k^b, \gamma_k^b \mid k \in \mathcal{T}_{\text{uni}}^b\}$ | $\hat{\boldsymbol{\theta}}_k^b = \boldsymbol{\theta}_{\text{pre}}^b + \gamma_k^b \cdot \boldsymbol{\tau}_{\text{uni}}^b \circ \boldsymbol{m}_k^b$ (reconstructed per-task $k$) | $\boldsymbol{\theta}_{\text{pre}}^b, \boldsymbol{\tau}_{\text{uni}}^b,$ $\{\boldsymbol{m}_k^b, \gamma_k^b \mid k \in \mathcal{T}_{\text{uni}}\}$ |

- **Similarity pre-computation:** We compute the pairwise cosine similarities between $M$ tasks in each block, across all $B$ blocks. Let the maximum size of any block task vector be $d_{\max}$, then the similarity computes takes $\mathcal{O}(d_{\max})$ per pair. With $\binom{M}{2}$ pairs per block, this step is $\mathcal{O}(BM^2 d_{\max})$.

- **Generating merging order:** Our greedy merging using Equation (1) is an instance of a specific form of clustering, called single-linkage clustering. We thus use the SLINK algorithm (Sibson, 1973) which takes as input the similarity matrix and generates a sorted list of merge orders for each block in $\mathcal{O}(M^2)$. For $B$ blocks, this takes $\mathcal{O}(BM^2)$. We then need to combine these per-block sorted lists, each of size $M - 1$, into a single global sorted list. Using a min-heap, this takes $\mathcal{O}(BM \log B)$ (Knuth, 1997). In total, this step takes $\mathcal{O}(BM^2 + BM \log B)$ and results in a global merge ordering across all blocks. Once the global merge ordering is obtained, we replay merges until the target size is met, simultaneously tracking task clusters via a DSU data structure, one per block. This gives the final groups $\{\mathcal{G}^b\}_{b=1}^B$ to which parameter merging is then applied. The per-block DSU overhead is $\mathcal{O}(M\alpha(M))$, where $\alpha(\cdot)$ is the Inverse Ackermann Function (Cormen et al., 2009). For all practical purposes, $\alpha(M) < 5$, resulting in $\approx \mathcal{O}(BM)$ for $B$ blocks.

- **Applying parameter merging:** We now merge parameters according to the final groupings in $\mathcal{G}^b$. For linear algorithms like TA, merging a group $g$ of size $|g|$ with task vectors of size $d$ takes $\mathcal{O}(|g|d)$. Summing over all groups within a block takes $\sum_{g \in \mathcal{G}^b} \mathcal{O}(|g| \cdot d) = \mathcal{O}((\sum_{g \in \mathcal{G}^b} |g|)d) = \mathcal{O}(Md)$. Repeating this for $B$ blocks and upper bounding $d$ with $d_{\max}$ results in $\mathcal{O}(BMd_{\max})$.

The total complexity is dominated by the similarity pre-computation (as $d_{\max}$ is typically larger than $B$), resulting in a final complexity of $\mathcal{O}(BM^2 d_{\max})$. Note however that $d_{\max}$ is much smaller than the total model dimension, at it only corresponds to the maximum size of any block of the model.

### 3.3 EXISTING MERGING METHODS IN COMBINATION WITH FLEXMERGE

FLEXMERGE provides the flexibility to choose any data-free merging algorithm $\mathcal{F}$ from a diverse set of existing approaches. Unlike traditional methods that operate at the level of full task vectors, FLEXMERGE applies merging algorithms at the block level, fusing block task vectors. We detail the exact block-level merging procedure for different algorithms next. In standard approaches like TA, TSV-M, and PCB-MERGING, task vectors are merged into a single unified task vector. When applied at the block-level, the merging outcome for any block $b$ can be denoted as: $\boldsymbol{\tau}_{\text{uni}}^b \leftarrow \mathcal{F}(\{\boldsymbol{\tau}_k^b \mid k \in \mathcal{T}_{\text{uni}}^b\})$ where $\mathcal{F}$ is the specific merging algorithm and $\mathcal{T}_{\text{uni}}^b$ is the subset of tasks for which the merging occurs. The final block parameters are then computed as $\boldsymbol{\theta}_{\text{uni}}^b = \boldsymbol{\theta}_{\text{pre}}^b + \boldsymbol{\tau}_{\text{uni}}^b$. Approaches such as CONSENSUS generate task-specific masks in addition to the unified vector: $\boldsymbol{\tau}_{\text{uni}}^b, \{\boldsymbol{m}_k^b \mid k \in \mathcal{T}_{\text{uni}}\} \leftarrow \mathcal{F}(\{\boldsymbol{\tau}_k^b \mid k \in \mathcal{T}_{\text{uni}}^b\})$. Then during inference, the task-specific weights for task $k$ are reconstructed as $\hat{\boldsymbol{\theta}}_k^b = \boldsymbol{\theta}_{\text{pre}}^b + \boldsymbol{\tau}_{\text{uni}}^b \circ \boldsymbol{m}_k^b$. CONSENSUS thus stores $\boldsymbol{\theta}_{\text{pre}}^b, \boldsymbol{\tau}_{\text{uni}}^b$, and the binary masks $\{\boldsymbol{m}_k^b \mid k \in \mathcal{T}_{\text{uni}}\}$ and defers per-task reconstruction to the inference time. This leads to a storage cost exceeding $2\times$ that of standard methods, which only store $\boldsymbol{\theta}_{\text{uni}}^b$. EMR-MERGING further generates task-specific scalars $\{\gamma_k^b \mid k \in \mathcal{T}_{\text{uni}}\}$ in addition to the masks, however the storage cost of these scalars is negligible. Table 1 summarizes the merging outcomes for different algorithms, applied at block-level within FLEXMERGE. Figure 7 (Section B) provides an illustrative diagram.

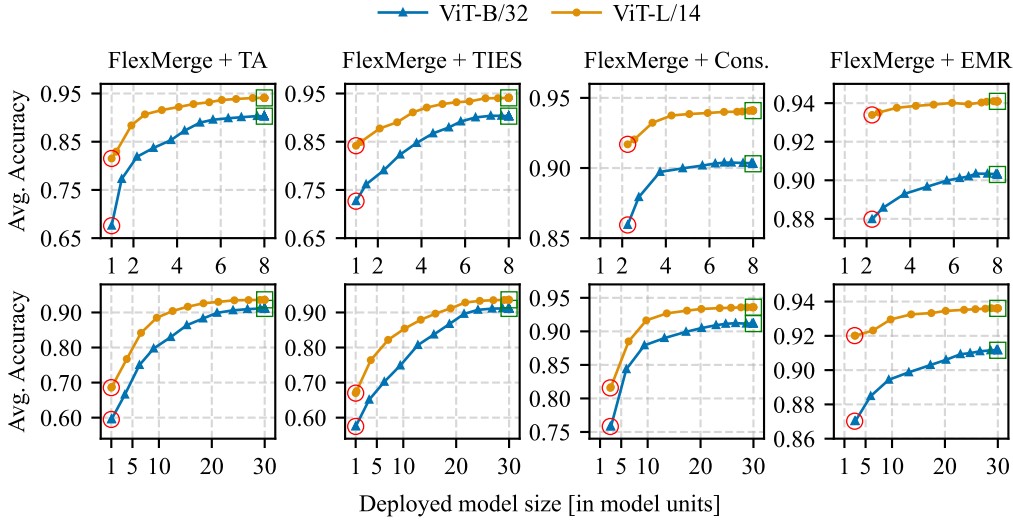

Figure 3: Merging 8 (top) and 30 (bottom) tasks. The accuracy-size trade-off shows rapid initial gains, followed by gradual improvement, reaching near fine-tuning accuracy well before the maximum size.

## 4 EXPERIMENTS

We split our evaluation as follows: *(i)* Merging on vision, PEFT and FFT benchmarks in Section 4.1; (ii) FLEXMERGE vs CHANNEL MERGING in Section 4.2; and *(iii)* ablation and efficiency analysis in Section 4.3. Lastly, multi-modal and OOD results are in Appendices C.4 and C.6.

**Merging algorithms.** We investigate the accuracy-size trade-off for several data-free merging algorithms including Vanilla Averaging, TA (Ilharco et al., 2023), TIES-MERGING (Yadav et al., 2023), PCB-MERGING (Du et al., 2024), TSV-M (Gargiulo et al., 2025), WUDI-MERGING (Cheng et al., 2025), CONSENSUS (Wang et al., 2024) and EMR-MERGING (Huang et al., 2024) on extensive vision and NLP benchmarks. As noted earlier, the focus of our work is data-free model merging. Hence, existing data-based algorithms such as SURGERY (Yang et al., 2024a), ADAMERGING (Yang et al., 2024b), TWIN-MERGING (Lu et al., 2024), *etc.* are not directly comparable in our setting.

**Hyperparameters.** For TA, we set $\lambda = 1.5$. For TIES-MERGING, we use a sparsity ratio of $0.1$ and employ the recommended value of $\lambda = 1$. For CONSENSUS, we set the hyperparameter responsible for controlling the amount of information extracted by masks to $0.6$ for all tasks and use TIES-MERGING as the algorithm to generate unified task vectors. For FLEXMERGE, we set the block granularity at the level of individual components within the transformer layer, *i.e.*, the attention, MLP, and layer normalization modules are treated as separate blocks during the merging process.

### 4.1 MERGING RESULTS

**Merging 8 and 30 vision models.** For the image classification tasks, we follow the setup from existing work (Huang et al., 2024; Yadav et al., 2023). Specifically, we use two versions of the CLIP model (Radford et al., 2021), incorporating ViT-B/32 and ViT-L/14 as visual encoders (Dosovitskiy et al., 2021). We evaluate on the standard 8 task benchmark (Ilharco et al., 2023) as well as an extended 30 task benchmark (detailed in Appendix B.2). Figure 3 plots average accuracy vs. deployed model size (in multiples of a single fine-tuned model). For FLEXMERGE + TA, the accuracy increases fairly rapidly as the model size grows beyond $1\times$. The gains are significant (top row), where the accuracy reaches $> 80\%$ at size $2\times$ from only $67.5\%$ at size $1\times$ for the ViT-B/32 model in the 8 task setup. Similar gains are also observed for 30 tasks (bottom row).

Masking-based approaches, CONSENSUS and EMR-MERGING, begin with substantially higher accuracy than TA and TIES-MERGING, but their smallest size exceeds $1\times$ due to the need to store pre-trained weights and binary masks (Section 3.3). On 8 tasks, CONSENSUS was shown to match

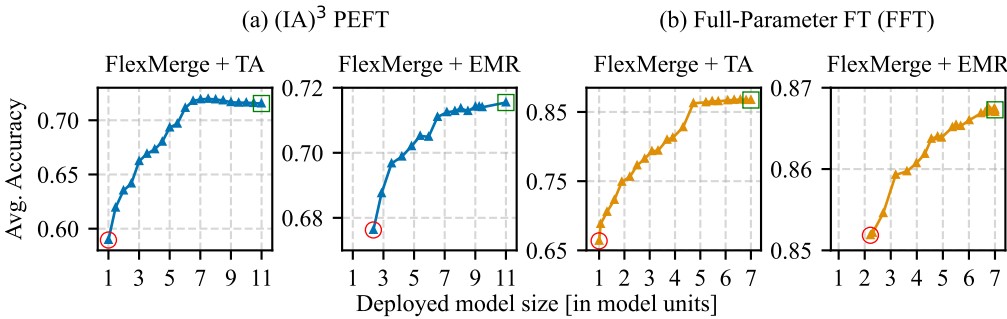

Figure 4: FLEXMERGE + TA gains 7.2% for (IA)$^3$ going from $1\times$ to $3\times$ and more than $9\%$ for FFT when just doubling the size from $1\times$ to $2\times$. EMR begins with higher accuracy, yet, substantially benefits from increased size.

fine-tuned accuracy at small sizes, but only when its extraction parameter is separately tuned per task (Wang et al., 2024). FLEXMERGE + CONSENSUS also shows strong gains, improving from $76\%$ at $\approx 3\times$ to $84.5\%$ at $\approx 6\times$ for ViT-B/32 in 30 tasks. EMR-MERGING maintains high accuracy even at the smallest size. Yet, it exhibits an accuracy gap w.r.t the fine-tuned models, which can be effectively reduced by increasing the deployed model size. Larger ViT-L/14 models achieve higher accuracy across all methods, but the accuracy-size trade-off remains similar: rapid initial gains followed by gradual improvements. Most algorithms approach the fine-tuning accuracy (denoted by □) well before maximum size, around $6\times$ for 8 tasks and $23.5\times$ for 30 tasks. Thus, in cases requiring storage of all fine-tuned models, FLEXMERGE can reduce size by about $25\%$ with little accuracy loss.

**Merging 11 PEFT models.** We adopt the experimental setup from prior work (Huang et al., 2024; Yadav et al., 2023). Specifically, we employ the (IA)$^3$ (Liu et al., 2022) PEFT method on the T0-3B (Sanh et al., 2022) base model using 11 diverse datasets sourced from (Yadav et al., 2023) (detailed in Section B.3). Figure 4(a) demonstrates the benefits of deploying larger model sizes, where in this case the model size is measured with respect to the (IA)$^3$ modules. FLEXMERGE + TA achieves notable gains, increasing accuracy from $59\%$ at size $1\times$ to $66.2\%$ at $3\times$, a $7.2\%$ improvement. Similarly, FLEXMERGE + EMR-MERGING surpasses $70\%$ accuracy at $5\times$, starting from $67.6\%$ at the lowest size of $2.34\times$. We observe similar trends for other algorithms, included in Appendix C.2.

**Merging 7 FFT models.** For this experiment, we closely follow the setup from prior work (Du et al., 2024; Yadav et al., 2023). We use T5-Base and T5-Large as base models, applying full-parameter fine-tuning on 7 datasets sourced from (Yadav et al., 2023) (detailed in Appendix B.4). Figure 4(b) illustrates the trade-off between model size and accuracy for the T5-Large model. Here, one unit of model size corresponds to the full size of a single model. FLEXMERGE + TA gains more than $9\%$ to reach an accuracy of $75\%$ when just doubling the size from $1\times$ to $2\times$. Similarly, FLEXMERGE + EMR-MERGING surpasses $86\%$ at size $4\times$, starting from $85.2\%$ at its lowest size of $2.2\times$. Consistent with our observations on vision tasks, FLEXMERGE + TA reaches very close to the fine-tuning accuracy around size $5\times$, much in advance of full size $7\times$. Thus, scaling the model size benefits both ends of the spectrum. Results for other combinations are included in Appendix C.3.

**Cross-algorithm analysis.** Thus far, we evaluated the accuracy-size trade-off per algorithm. We now compare algorithms at same size, yielding two interesting findings: *(i)* the performance gap between different algorithms significantly narrows at slightly larger sizes; and *(ii)* the algorithms rankings also alter in many cases, with simpler algorithms rivaling or surpassing advanced ones. This behavior aligns with the phenomenon we observed earlier: as scale increases, parameter interference between tasks reduces substantially, leading to greater natural parameter disentanglement and higher accuracy. Consequently, the explicit interference-reduction mechanisms built into advanced approaches such as TRIM in TIES or competition balancing in PCB offer marginal added benefit, because much of the interference is already mitigated simply by the increase in scale. Thus simple algorithms begin rivaling their more sophisticated counterparts at larger sizes. In Figure 5 on vision tasks,

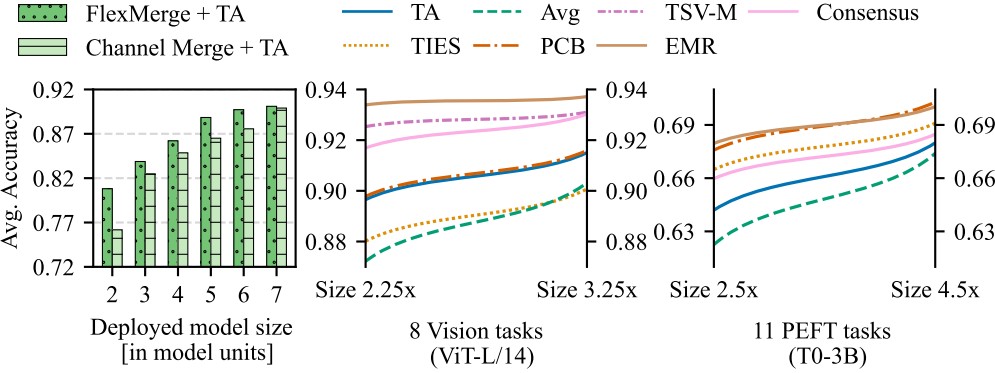

Figure 5: (Left) FLEXMERGE + TA outperforms CHANNEL MERGING + TA across all sizes. (Center, Right) Algorithm rankings shift even at modestly larger sizes, with simpler methods rivaling advanced ones. We show sizes just over CONSENSUS and EMR-MERGING's lowest size for a wholistic comparison.

vanilla averaging exceeds TIES-MERGING at size $3.25\times$ while TA overlaps with PCB. While EMR-MERGING and CONSENSUS stay atop on vision, they are surpassed by PCB on PEFT at size $4.5\times$. Crucially, all algorithms remain within $3 - 4\%$ on both benchmarks at increased sizes despite originating with a much larger gap at size $1\times$ (see Figure 1(b)). *Our findings provide encouraging evidence to develop and compare algorithms at sizes $> 1\times$ rather than only at $1\times$.*

## 4.2 FLEXMERGE VS CHANNEL MERGING

CHANNEL MERGING (Zhang et al., 2025) uses K-Means clustering per layer, following a fixed same value of $K$ for every layer. Each choice of $K \in \{2, 3, \dots, M - 1\}$ results in a merged model of the corresponding size. Figure 5 charts the average accuracy with TA and ViT-B/32 for a set of integer model sizes, excluding the extremes $1\times$ and $8\times$ where both approaches have the identical accuracy. Recall that CHANNEL MERGING does not support fractional sizes. FLEXMERGE achieves higher accuracy than CHANNEL MERGING in all cases, thanks to its greedy pairwise merging approach which allows flexible number of groups per layer instead of restrictive clustering. Results with TIES-MERGING and visualization of clusters is included in Appendix C.5.

## 4.3 ANALYSIS

**Ablations on the merging procedure.** We ablate on the similarity functions (min, max, average, comparing unified vectors) for Equation (1) and merging orders (left-to-right, right-to-left, greedy) in FLEXMERGE using the ViT-B/32 model on $8$ tasks. We also investigate random block selection over cosine similarity. Figure 6 shows that the min strategy performs the best, though other strategies are also competitive. For merging order, right to left performs the worst as expected since the final layers in neural networks tend to be more specialized and merging them first hurts accuracy. While left to right seems ideal, it can be too strict and therefore greedy emerges as the best. We further analyze the merging order of greedy in Appendix C.10. Random selection is competitive, but generally underperforms when compared across algorithm. Based on these findings, we set FLEXMERGE to use greedy with cosine similarity (min strategy) by default. For more ablations, see Appendix C.8.

**Merging and inference efficiency.** Table 2 shows that FLEXMERGE achieves highly efficient data-free merging, generating all deployed sizes in about $20\,\mathrm{sec}$ for up to 30 tasks. For inference with FLEXMERGE, each request follows a unique forward path through the merged model using task-specific blocks (Figure 1(a)). For a model of size $1\times$, all tasks share a single path, but the classification heads are always applied separately. We load the tensors of merged model (size $> 1\times$) into the GPU memory once and create $M$ task-specific model views that reference these shared tensors to process task batches in parallel. Standard merging, by contrast, processes all tasks in a single batch before splitting for task-specific heads. We simulate the worst case arrival, where

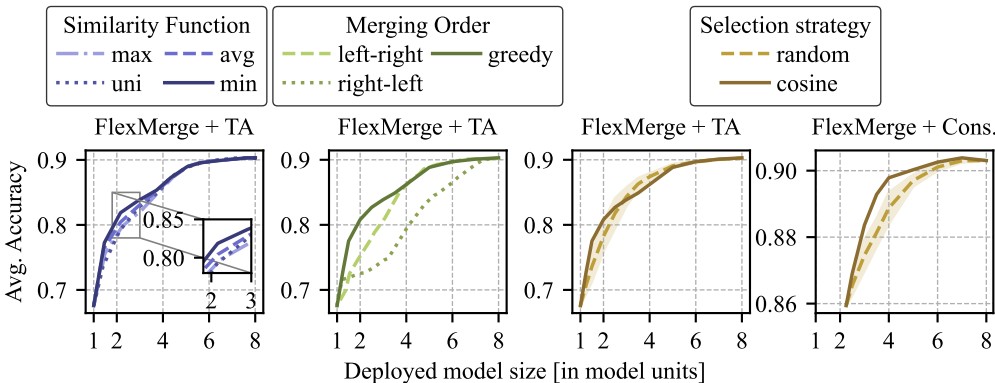

Figure 6: Ablation results for FLEXMERGE reveal that the min similarity strategy and greedy merging perform the best, while cosine similarity generally outperforms random selection.

inference batches corresponding to all tasks arrive at once. We consider 50 consecutive batches of size 256 (totaling 12800 samples). Each batch contains 32 samples per task across 8 tasks. Table 3 shows that FLEXMERGE maintains inference speed comparable to standard merging for both ViT-B/32 and ViT-L/14, demonstrating that larger models can enhance accuracy without slowing inference.

Table 2: FLEXMERGE's merging time.

| Method | ViT-B/32 (s) | ViT-L/14 (s) |
|---|---|---|
| *8 Tasks* | | |
| TA (size 1×) | ≈ 0.8 | ≈ 2.6 |
| FLEXMERGE (all sizes) | ≈ 2.3 | ≈ 3.4 |
| *30 Tasks* | | |
| TA (size 1×) | ≈ 1.9 | ≈ 6.1 |
| FLEXMERGE (all sizes) | ≈ 20 | ≈ 31 |

Table 3: Comparing inference time of FLEXMERGE against standard model merging. The overheads are negligible.

| Model | Algorithm | Size | Inference Cost (/12800 items) |
|---|---|---|---|
| ViT-B-32 | Standard Merging | 1× | $12.30 \pm 0.21$ ms |
| | FLEXMERGE | > 1× | $12.21 \pm 0.41$ ms |
| ViT-L-14 | Standard Merging | 1× | $118.70 \pm 1.78$ ms |
| | FLEXMERGE | > 1× | $120.53 \pm 0.32$ ms |

## 5 DISCUSSION AND CONCLUSION

**Benefits.** Different merging algorithms have different advantages: EMR and CONSENSUS achieve high accuracy but require task-specific reconstruction during inference, incurring overheads. FLEXMERGE can also mitigate this overhead as larger deployed models need fewer blocks to be reconstructed (see Appendix C.9). In contrast, TIES and TA avoid reconstruction but have lower accuracy. FLEXMERGE provides flexibility, letting practitioners choose algorithms and balance accuracy, reconstruction overhead, and model size for various deployment scenarios.

**Limitations.** Most works, including FLEXMERGE, are limited to merging models with the same architecture as merging heterogeneous models remains challenging (Singh & Jaggi, 2020; Imfeld et al., 2024). Secondly, the theoretical insights for effective model merging are limited (Ortiz-Jimenez et al., 2023). For FLEXMERGE, how to obtain the optimal merged model for any given size remains unclear. Although extensive ablations help guide (Section 4.3), further investigation is needed to understand the bounds of the accuracy-size trade-off.

We introduced FLEXMERGE, a flexible, data-free model merging framework that extends beyond traditional single-model fusion and offers precise control over fused model size. Extensive experiments show that the accuracy-size trade-off exhibits favorable properties for several algorithms, benefiting from rapid accuracy gains with modest size increments. Future work may explore specialized algorithms for block-level merging.

ACKNOWLEDGMENTS

Akash was supported by the EPFL Doc.Mobility fellowship during his research visit to Carnegie Mellon University. This work was also partially supported by the NSF grants CCF 2045694, CCF 2428569, CNS-2112471, CPS-2111751, ONR grant N00014-23-1-2149, AI2C Seed grant, and by the Swiss National Science Foundation, under the project "FRIDAY: Frugal, Privacy-Aware and Practical Decentralized Learning", SNSF proposal No. 10.001.796. The authors are thankful to Martijn de Vos and Arian Raje for their helpful feedback during the preparation of this work.

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

ORGANIZATION OF THE APPENDIX

## A  EXTENDED RELATED WORK

Table 4 characterizes different merging algorithms on their data-based/data-free nature and merged model size. Below we discuss additional related work.

**Multi-Task Learning (MTL).** The traditional approach to obtaining a model with multi-task capabilities is MTL which trains a single model using training data from multiple tasks together (Vandenhende et al., 2021; Sanh et al., 2022). However, MTL suffers from not only (i) the expensive computational cost for training, but also (ii) the limited data availability due to data privacy (Jin et al., 2023; Yang et al., 2024b). In comparison, model merging bypasses these challenges by combining the fine-tuned model weights directly, without training data, thus offering a more cost-effective approach to building a multi-task model. Existing research in MTL also focuses on grouping tasks *i.e.*, identifying subsets of task that derive positive benefit from training together (Standley et al., 2020; Fifty et al., 2021). Since training one model for all tasks can lead to suboptimal performance due to task conflicts and competition for model capacity, these methods train separate multi-task models for specific task groups. This can be seen as conceptually similar to our approach of having different task subsets per merged block to improve performance (see Figure 1(a)). While our approach performs grouping during merging, these approaches perform grouping during training.

**Model Merging.** Besides the works discussed in Section 2, recent works also focus on merging models fine-tuned specifically with Low-Rank Adaptation (LoRA) (Stoica et al., 2025; Zhao et al., 2025; Tang et al., 2024b). Other efforts focus on developing approaches for fine-tuning that result in lower interference during downstream merging (Lee et al., 2025; Jin et al., 2025; Ortiz-Jimenez et al., 2023). LiNeS (Wang et al., 2025) proposes a post-training editing technique to reduces negative interference between parameters by scaling parameter updates based on their layer depth. FLEXMERGE can seamlessly incorporate these recent advances.

Table 4: Comparison of merging algorithms by data dependency and merged model size.

| Algorithm | Data Free | Size | Storage beyond the unified model |
|---|---|---|---|
| Weight Average | ✓ | $1\times$ (Fixed) | – |
| TA (Ilharco et al., 2023) | ✓ | $1\times$ (Fixed) | – |
| TIES-MERGING (Yadav et al., 2023) | ✓ | $1\times$ (Fixed) | – |
| PCB-MERGING (Du et al., 2024) | ✓ | $1\times$ (Fixed) | – |
| CONSENSUS (Wang et al., 2024) | ✓ | $>2\times$ (Fixed) | Stores masks and $\theta_{\text{pre}}$ |
| EMR-MERGING (Huang et al., 2024) | ✓ | $>2\times$ (Fixed) | Stores masks and $\theta_{\text{pre}}$ |
| TSV-M (Gargiulo et al., 2025) | ✓ | $1\times$ (Fixed) | – |
| WUDI-MERGING (Cheng et al., 2025) | ✓ | $1\times$ (Fixed) | – |
| REGMEAN (Jin et al., 2023) | ✗ | $1\times$ (Fixed) | – |
| FISHER MERGING (Matena & Raffel, 2022) | ✗ | $1\times$ (Fixed) | – |
| ADAMERGING (Yang et al., 2024b) | ✗ | $1\times$ (Fixed) | – |
| SURGERY (Yang et al., 2024a) | ✗ | $>1\times$ (Fixed) | Stores task-specific adapters |
| WEMOE (Tang et al., 2024a) | ✗ | $\gg 1\times$ (Fixed) | Stores MLP modules for all tasks |
| TWIN-MERGING (Lu et al., 2024) | ✗ | $>2\times$ (Fixed) | Stores compressed diff. vectors and $\theta_{\text{pre}}$ |

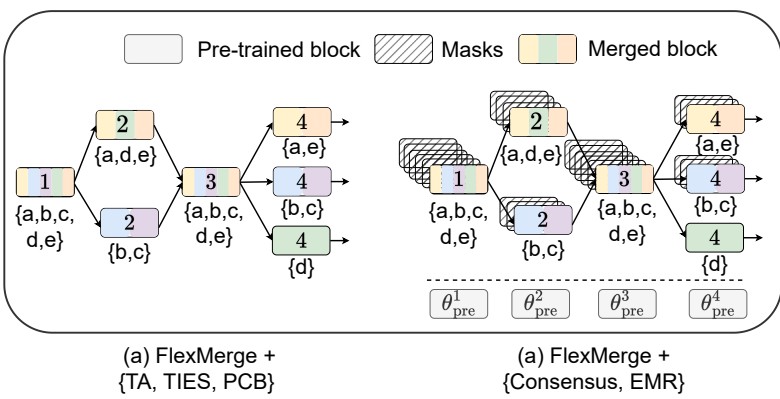

(a) FlexMerge +
{TA, TIES, PCB}

(a) FlexMerge +
{Consensus, EMR}

Figure 7: FLEXMERGE in combination with different merging algorithms. Standard methods such as TA, TIES-MERGING, *etc*. generate the merged parameters per block. Recent methods such as CONSENSUS and EMR-MERGING generate binary masks in addition to the unified parameters. They also store the pre-trained parameters for task-specific reconstruction.

## B  ADDITIONAL DETAILS

### B.1  HOW TO APPLY EXISTING MERGING ALGORITHMS AT THE BLOCK-LEVEL?

While most merging algorithms can be directly applied at the block-level, we list them below and explain any specific adaptations that help improve performance. Figure 7 provides an illustrative diagram for FLEXMERGE in combination with different merging algorithms.

- **Averaging**. Applied directly to obtain $\boldsymbol{\tau}_{\text{uni}}^{b}$.

$$\mathcal{F}(\{\boldsymbol{\tau}_k^b \mid k \in \mathcal{T}_{\text{uni}}^b\}) = \frac{1}{|\mathcal{T}_{\text{uni}}^b|} \sum_{k \in \mathcal{T}_{\text{uni}}^b} \boldsymbol{\tau}_k^b$$

- **TA** (Ilharco et al., 2023). Applied directly to obtain $\boldsymbol{\tau}_{\text{uni}}$.

$$\mathcal{F}(\{\boldsymbol{\tau}_k^b \mid k \in \mathcal{T}_{\text{uni}}^b\}) = \lambda \cdot \frac{1}{|\mathcal{T}_{\text{uni}}^b|} \sum_{k \in \mathcal{T}_{\text{uni}}^b} \boldsymbol{\tau}_k^b$$

For all experiments with TA, we use $\lambda = 1.5$. Note that our definition of $\lambda$ for TA excludes the $1/|\mathcal{T}_{\text{uni}}^b|$ factor (see Section 3.1), in contrast to prior work. This modification is essential for ensuring reasonable merging performance with TA, as the number of tasks being fused per block varies throughout the bottom-up fusion process, ranging between 2 and $M$.

- **TIES-MERGING** (Yadav et al., 2023). We apply trimming to the full task vectors $\boldsymbol{\tau}_k, k \in [M]$ to retain the top $10\%$ of the parameters before beginning the greedy merging process. This is because selecting the top parameters globally performs better than selecting them within each block. During the bottom-up merging process, only the elect sign and disjoint merge steps of TIES-MERGING are performed. Lastly, we employ the recommended value of $\lambda = 1$.

- **PCB-MERGING** (Du et al., 2024). Applied directly at the block-level by executing the steps of intra-balancing, inter-balancing and drop and rescale at the block-level. In PCB-MERGING, the drop operation for any task vector depends on other task vectors that it is being merged with. This contrasts with TIES-MERGING where the drop (or trim) operation is solely dependent on the magnitude of values in the task vector. Therefore, global trimming is not possible in PCB-MERGING and we execute it block-wise. We set the sparsity ratio to $10\%$ for vision tasks and $20\%$ for PEFT and FFT experiments. Additionally, we employ the recommended value of $\lambda = 1$ across all experiments.

- **CONSENSUS** (Wang et al., 2024). Applied directly wherein the merging results in not only the unified task vector but also the task specific masks.

$$\boldsymbol{\tau}_{\text{uni}}^b, \{\boldsymbol{m}_k^b \mid k \in \mathcal{T}_{\text{uni}}^b\} \leftarrow \mathcal{F}(\{\boldsymbol{\tau}_k^b \mid k \in \mathcal{T}_{\text{uni}}^b\})$$

The task-specific weights for task $k \in \mathcal{T}_{\text{uni}}^b$ corresponding to this merged block $b$ are then reconstructed during the reconstruction process as:

$$\hat{\boldsymbol{\theta}}_k^b = \boldsymbol{\theta}_{\text{pre}}^b + \boldsymbol{\tau}_{\text{uni}}^b \circ \boldsymbol{m}_k^b$$

We use TIES-MERGING as the algorithm to generate $\boldsymbol{\tau}_{\text{uni}}^b$ within CONSENSUS. Note that the above version of CONSENSUS corresponds to the compression application in (Wang et al., 2024). The alternative version of CONSENSUS that corresponds to merging using masks can be also directly leveraged within FLEXMERGE.

- **EMR-MERGING** (Huang et al., 2024). Applied directly wherein the merging results in the unified task vector, the task specific masks and task-specific rescalers.

$$\boldsymbol{\tau}_{\text{uni}}^b, \{\boldsymbol{m}_k^b, \gamma_k^b \mid k \in \mathcal{T}_{\text{uni}}^b\} \leftarrow \mathcal{F}(\{\boldsymbol{\tau}_k^b \mid k \in \mathcal{T}_{\text{uni}}^b\})$$

The task-specific weights for task $k \in \mathcal{T}_{\text{uni}}^b$ corresponding to this merged block $b$ are then reconstructed during the reconstruction process as:

$$\hat{\boldsymbol{\theta}}_k^b = \boldsymbol{\theta}_{\text{pre}}^b + \gamma_k^b \cdot \boldsymbol{\tau}_{\text{uni}}^b \circ \boldsymbol{m}_k^b$$

- **TSV-M** (Gargiulo et al., 2025). Applied directly per-block as the method originally also operates layerwise. 2D parameters are merged using their task singular vectors while 1D parameters just use averaging for merging as done in their original work.

- **WUDI-MERGING** (Cheng et al., 2025). Applied directly per-block as the method originally also operates layerwise. We use the same learning rate of $10^{-5}$ as used by the authors within the method. Lastly, we set the number of iterations to 300 for 8 tasks and 1000 for 30 tasks.

## B.2 VISION BENCHMARK

The 8 task vision benchmark (Ilharco et al., 2023) comprises the following datasets: 1. SUN397 (Xiao et al., 2010), 2. Cars (Krause et al., 2013), 3. RESISC45 (Cheng et al., 2017), 4. EuroSAT (Helber et al., 2019), 5. SVHN (Yuval, 2011), 6. GTSRB (Stallkamp et al., 2011), 7. MNIST (LeCun, 1998), and 8. DTD (Cimpoi et al., 2014). We extend the 8 task vision benchmark with 12 additional datasets sourced from (Wang et al., 2024), including: 9. CIFAR100 (Krizhevsky et al., 2009), 10. STL10 (Coates et al., 2011), 11. Flowers102 (Nilsback & Zisserman, 2008), 12. OxfordIIITPet (Parkhi et al., 2012), 13. PCAM (Veeling et al., 2018), 14. FER2013 (Goodfellow et al., 2013), 15. EMNIST (Cohen et al., 2017), 16. CIFAR10 (Krizhevsky et al., 2009), 17. Food101 (Bossard et al., 2014), 18. FashionMNIST (Xiao et al., 2017), 19. RenderedSST2 (Socher et al., 2013; Radford et al., 2019) and 20. KMNIST (Clanuwat et al., 2018). The remaining 10 datasets for our 30 task benchmark are sourced from (Huang et al., 2024), which include: 21. Weather (Xiao et al., 2021), 22. Vegetables (Ahmed et al., 2021), 23. MangoLeafBD (Ahmed et al., 2023), 24. Landscape Recognition (DeepNets), 25. Beans (Lab, 2020), 26. Intel Images (Bansal, 2019), 27. Garbage Classification (CCHANG, 2018) 28. Kvasir (Pogorelov et al., 2017), 29. KenyanFood13 (Jalal et al., 2019) and 30. Dogs (Khosla et al., 2011)

### B.3 Parameter efficient fine-tuning (PEFT)

We fine-tune (IA)[3] modules on 11 diverse datasets, including RTE (Giampiccolo et al., 2007), CB (De Marneffe et al., 2019), Winogrande (Sakaguchi et al., 2021), WiC (Pilehvar & Camacho-Collados, 2018), WSC (Levesque et al., 2012), COPA (Roemmele et al., 2011), H-SWAG (Zellers et al., 2019), Story Cloze (Sharma et al., 2018), and ANLI (Nie et al., 2019) (R1 to R3). In addition, we leverage prompt templates from the Public Pool of Prompts (P3) (Bach et al., 2022) which convert each dataset example into a text-to-text format, where each label is mapped to a unique string representation. We report the median performance across all templates for each dataset. Evaluating multiple prompt templates increases the evaluation runtime significantly. To ensure the runtime remains manageable, we cap the maximum number of test samples at 1000 per dataset.

### B.4 Full parameter fine-tuning

The 7 datasets that we consider for fine-tuning include: PAWS (Zhang et al., 2019), QASC (Khot et al., 2020), QuaRTz (Tafjord et al., 2019), Story Cloze (Sharma et al., 2018), WikiQA (Yang et al., 2015), Winogrande (Sakaguchi et al., 2021), and WSC (Levesque et al., 2012). During training and evaluation, we apply a specific prompt template from P3 (Bach et al., 2022) to each dataset. Each model is trained for up to 75 000 optimization steps, with early stopping if validation accuracy does not improve over five consecutive evaluation rounds. Performance is evaluated every 5 steps for WSC and every 100 steps for other datasets, using the full validation set. Training is conducted using the Adam optimizer with a constant learning rate of 0.0001 and an effective batch size of 1024. The maximum sequence length is set to 128, and bfloat16 precision is used for both training and evaluation.

### B.5 Multi-modal benchmark

We evaluate the performance of merging fine-tuned checkpoints of BEiT-3-base model (Wang et al., 2022) under FLEXMERGE. We consider four datasets: COCO Captioning (Lin et al., 2014) (Image Captioning), ImageNet-1k (Deng et al., 2009) (Image Classification), NLVR2 (Suhr et al., 2019) (Visual Reasoning) and COCO Retrieval (Lin et al., 2014) (Image-Text Retrieval). The individual checkpoints, each fine-tuned on one of these datasets, are available in UniLM repository[3]. We merge layers that are common to all checkpoints *i.e.*, excluding the task-specific classification heads. We report accuracy for ImageNet-1k, NLVR2 and COCO Retrieval, meanwhile for COCO Captioning, we report BLEU-4 (Papineni et al., 2002), CIDEr (Vedantam et al., 2015), ROUGE-L (Lin, 2004) and METEOR (Banerjee & Lavie, 2005).

## C Additional results

In this section, we provide the remaining results for each benchmark.

### C.1 Vision benchmark

Figures 8 and 9 show the results on 8 and 30 tasks respectively, for Averaging, PCB-MERGING, TSV-M and WUDI-MERGING extending our results in Figure 3. FLEXMERGE + Avg. gains over $15\%$ in accuracy by just doubling the size from $1\times$ to $2\times$ in the 8-task benchmark under the ViT-B/32 model. PCB-MERGING starts with higher accuracy at size $1\times$ than vanilla averaging, thanks to its competition balancing procedure. Yet, it significantly benefits from increasing the deployed model size. Under the same setup as above, FLEXMERGE + PCB-MERGING gains $7.5\%$, by just doubling the size from $1\times$ to $2\times$. TSV-M and WUDI-MERGING significantly lead both prior algorithms at size $1\times$, still showing steep improvements of over $4\%$ from size $1\times$ to $2\times$. We observe similar steep initial gains for both the algorithms even with 30 tasks. Notably, in all cases, we observe that a near fine-tuning accuracy is reached well in advance of the maximum size, around $6\times$ for 8 tasks and around $23.5\times$ for 30 tasks.

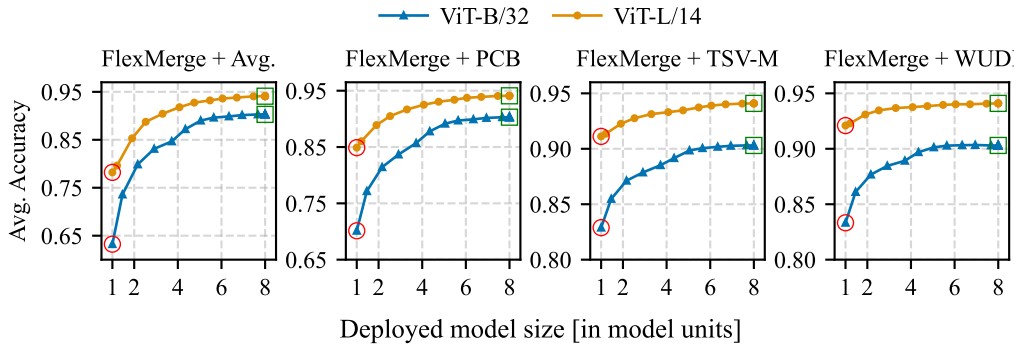

Figure 8: Accuracy-size trade-off for 8 tasks.

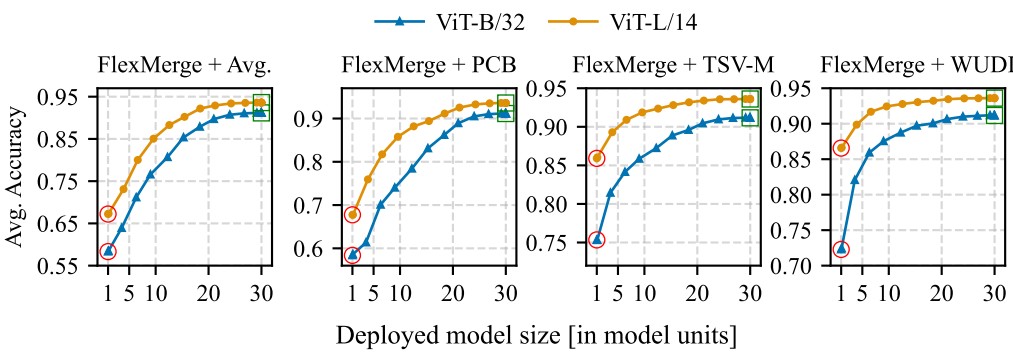

Figure 9: Accuracy-size trade-off for 30 tasks.

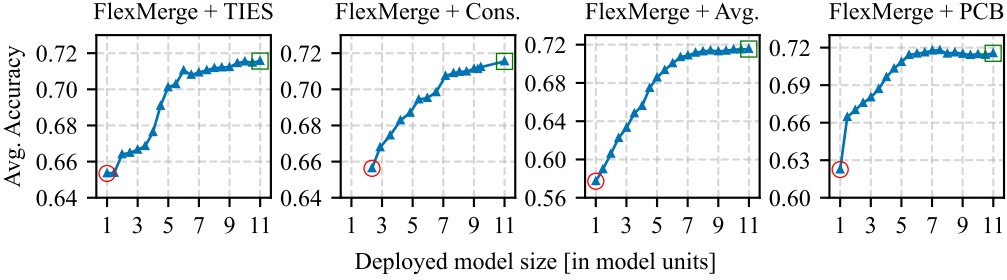

Figure 10: Remaining combinations on the (IA)$^3$ PEFT benchmark with **T0-3B**.

## C.2 (IA)$^3$ PEFT BENCHMARK

We extend the results of Figure 4 with four additional algorithms in Figure 10. All algorithms demonstrate a steep initial rise in accuracy, except TIES-MERGING for which the gains appear sharply after a steady initial rise. Notably, at size $7\times$, all algorithms are within $1\%$ of the fine-tuning accuracy (denoted by □). Therefore, practitioners aiming to deploy all 11 fine-tuned models for high accuracy can be benefit from a reduction of 4 model units without losing too much accuracy.

---

[3] https://github.com/microsoft/unilm

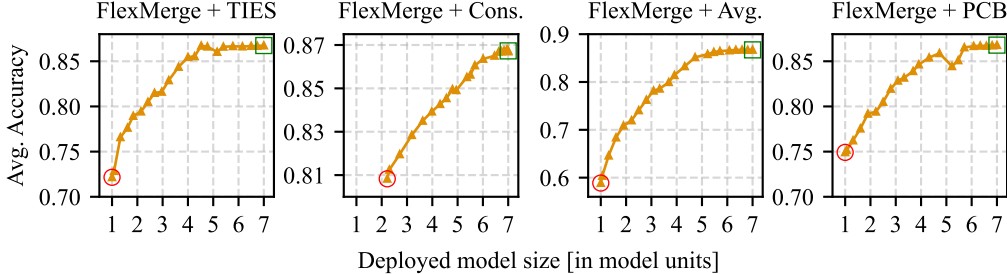

Figure 11: Remaining combinations on the FFT benchmark with **T5-Large**.

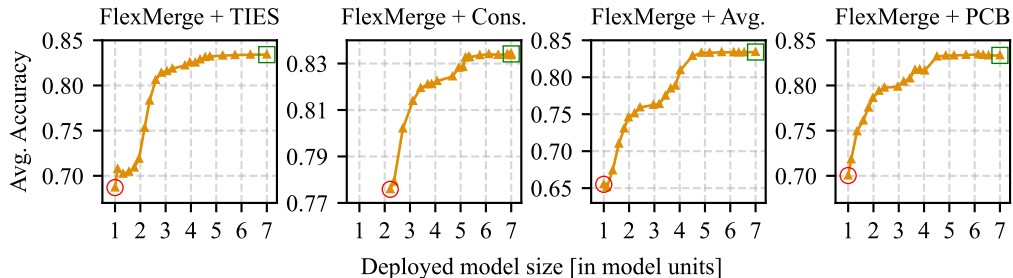

Figure 12: Remaining combinations on the FFT benchmark with **T5-Base**.

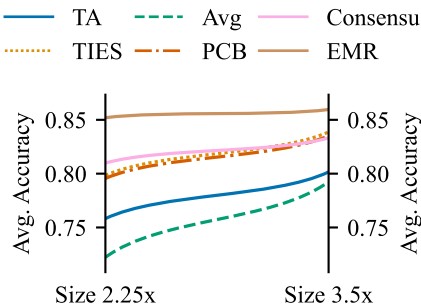

Figure 13: Results on T5-Large. Algorithms rankings are not consistent across sizes. In particular, PCB-MERGING and TIES-MERGING surpass the accuracy of masking-based approach CONSENSUS while averaging almost attains the accuracy of TA at size $3.5\times$.

### C.3 FULL-PARAMETER FINE-TUNING (FFT) BENCHMARK

We extend the results of Figure 4 with four additional merging algorithms on two models. Figure 11 and Figure 12 chart the results for T5-Large and T5-Base respectively. Just doubling the deployed model size from $1\times$ to $2\times$ significantly improves the accuracy across all combinations. We note gains of more than $7\%$, $12\%$ and $4.3\%$ for TIES-MERGING, Averaging and PCB-MERGING respectively under the T5-Large model. Similarly, we note gains of nearly $9\%$ for both Averaging and PCB-MERGING under the T5-Base model. TIES-MERGING under the T5-Base model demonstrates a sharp rise around size $2\times$, reaching $80.6\%$ accuracy at just $2.7\times$ from an accuracy of $68.7\%$ at size $1\times$, a gain of nearly $12\%$. Except for FLEXMERGE + CONSENSUS under T5-Large which demonstrates a linear trade-off, all other combinations exhibit a highly favorable trade-off, validating the benefits of our approach across diverse scenarios. Lastly, we also observe that algorithm rankings remain inconsistent on the FFT benchmark as well when sizes are increased, as illustrated in Figure 13.

## C.4    MULTI-MODAL BENCHMARK

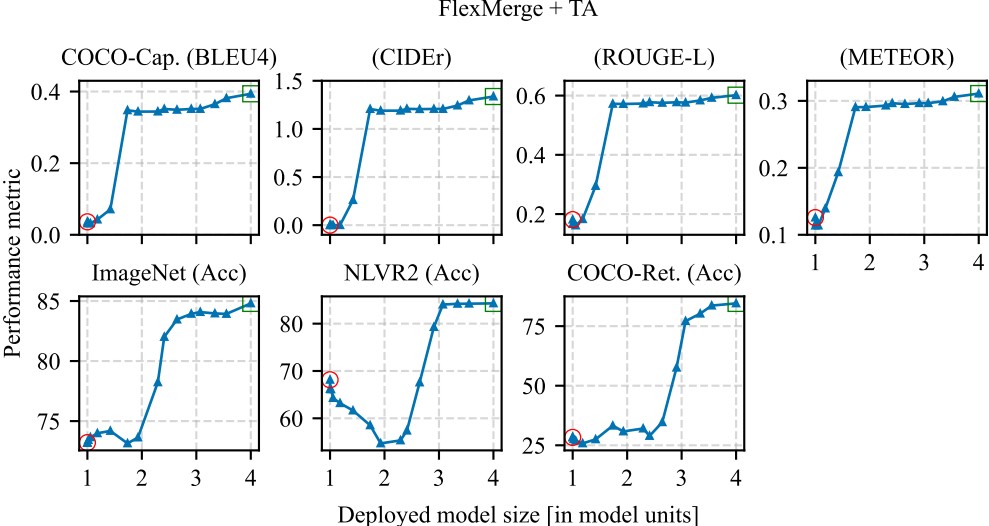

Figure 14: Accuracy-size trade-off for FLEXMERGE + TA on the multi-modal benchmark with **BEiT-3-base**. The performance metrics for each dataset are shown in parentheses.

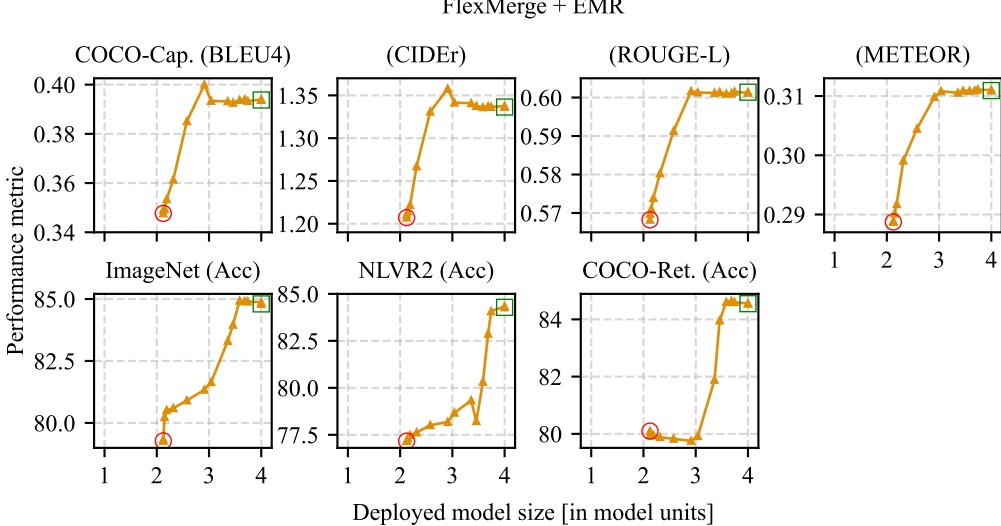

Figure 15: Accuracy-size trade-off for FLEXMERGE + EMR-MERGING on the multi-modal benchmark with **BEiT-3-base**. The performance metrics for each dataset are shown in parentheses.

In this section, we present the accuracy-size trade-off for the multi-modal benchmark detailed in Section B.5. Figures 14 and 15 chart the results for FLEXMERGE + TA and FLEXMERGE + EMR-MERGING respectively. The performance metrics differ across datasets, therefore we show each dataset separately. The first row in each figure corresponds to the COCO-Captioning dataset with 4 metrics while the second row corresponds to ImageNet, NLVR2 and COCO-Retrieval datasets, each evaluated with accuracy. For all metrics, higher values indicate better performance.

Across all datasets, we observe a significant gap between the performance at lowest merged size and the fine-tuned model performance. FLEXMERGE + TA shows significant benefit of increased size on the COCO-Captioning dataset where the performance steeply grows across all four metrics between

size $1\times$ and $2\times$. For ImageNet, NLVR2 and COCO-Retrieval the performance improves sharply between size $2\times$ and $3\times$. Interestingly, NLVR2 shows a drop in performance before rising sharply, indicating that merging with related tasks can provide complementary benefits. EMR-MERGING has a lowest size of over $2\times$ due to the cost of storing the pre-trained model and the masks. However, this provides a significantly higher starting performance than TA. The performance improvements for FLEXMERGE + EMR-MERGING are similar to FLEXMERGE + TA, where COCO-Captioning demonstrates a sharp rise between $2\times$ and $3\times$ while the remaining datasets demonstrate a sharp rise between $3\times$ and $4\times$. Thus, larger sizes can confer beneficial improvements across both algorithms, obviating the need to deploy all fine-tuned models for high performance.

## C.5 FLEXMERGE VS CHANNEL MERGING

In this section, we compare FLEXMERGE with CHANNEL MERGING under the TIES-MERGING algorithm, following the same setup as our experiments with TA in Figure 5. As shown in Figure 16, FLEXMERGE consistently achieves significantly higher accuracy than CHANNEL MERGING across all sizes. This is because CHANNEL MERGING enforces a fixed number of clusters per block, whereas FLEXMERGE allows vastly different number of clusters per layer by design. This flexibility, enabled by iterative pairwise merging, leads to substantial accuracy improvements. To better understand this effect in FLEXMERGE, we visualize the number of clusters per block in Figure 17, corresponding to merged model sizes of $2.16\times$. For clarity, we focus on Attention and MLP blocks, which constitute the bulk of the model size, omitting smaller blocks such as layer norms. In Figure 17, we observe that the number of clusters per block varies significantly—from 1 to 4—whereas CHANNEL MERGING, at a similar model size, would enforce exactly 2 clusters per block, limiting its effectiveness.

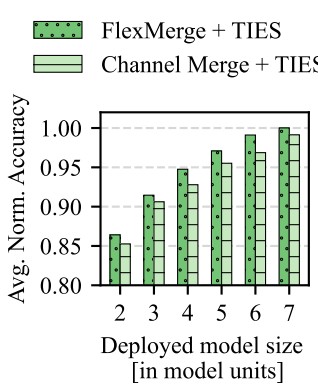

Figure 16: FLEXMERGE achieves higher accuracy than CHANNEL MERGING across all sizes.

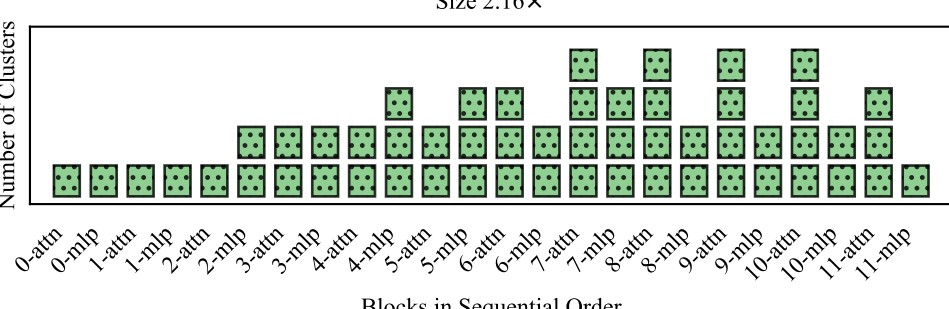

Figure 17: Visualizing the number of clusters per block in FLEXMERGE model of size $2.16\times$. The number of clusters vary significantly across blocks, ranging from 1 to 4. In contrast, for a model of size $2\times$, CHANNEL MERGING will have exactly 2 clusters in each block. This flexibility enables greater accuracy for FLEXMERGE over CHANNEL MERGING.

## C.6 OUT-OF-DISTRIBUTION (OOD) PERFORMANCE OF FLEXMERGE

To assess the OOD performance of FLEXMERGE, we conducted additional experiments on 6 OOD tasks using models merged from 8 in-domain tasks. For each OOD task, the prediction is obtained by ensembling the outputs of the 8 in-domain branches of the merged model (see Figure 1(a)).

**Setup:** In-domain tasks are SVHN, Cars, RESISC45, EuroSAT, SUN397, GTSRB, MNIST and DTD. Out-of-domain tasks are Weather, PCAM, Flowers102, Landscapes, Beans and Food101. We consider TIES as the merging algorithm as it known for its good performance OOD.

**Results.** Table 5 presents the average accuracy of each in-domain expert on the 6 OOD tasks. Table 6 presents the average accuracy of FLEXMERGE's merged model for varying sizes on the 6 OOD tasks. We observe that across all model sizes $> 1\times$, FLEXMERGE consistently outperforms the best single in-domain expert on these OOD tasks, suggesting that our method not only preserves but can also improve OOD generalization.

Table 5: Average OOD Accuracy of Individual In-domain Experts. Bold-faced values indicate the best in-domain expert.

| Model ↓ \| Dataset → | SVHN | Cars | RESISC45 | EuroSAT | SUN397 | GTSRB | MNIST | DTD |
|---|---|---|---|---|---|---|---|---|
| ViT-B-32 | 0.5538 | 0.6230 | 0.6268 | 0.5768 | **0.6366** | 0.5083 | 0.5060 | 0.5910 |
| ViT-L-14 | 0.7259 | 0.7152 | 0.6908 | 0.6797 | 0.7096 | **0.7328** | 0.7298 | 0.7193 |

Table 6: Average OOD Accuracy of FlexMerge at Varying Sizes. Bold-faced values indicate higher accuracy than the best in-domain expert in Table 5.

| Model ↓ \| Merged Size → | 1x | 1.5x | 2x | 2.5x | 3x | 4x | 6x | 8x |
|---|---|---|---|---|---|---|---|---|
| ViT-B-32 | 0.6289 | **0.6410** | **0.6475** | **0.6539** | **0.6567** | **0.6586** | **0.6548** | **0.6595** |
| ViT-L-14 | **0.7390** | **0.7394** | **0.7398** | **0.7403** | **0.7416** | **0.7453** | **0.7464** | **0.7488** |

## C.7 SCALING LAWS FOR FLEXIBLE MODEL MERGING

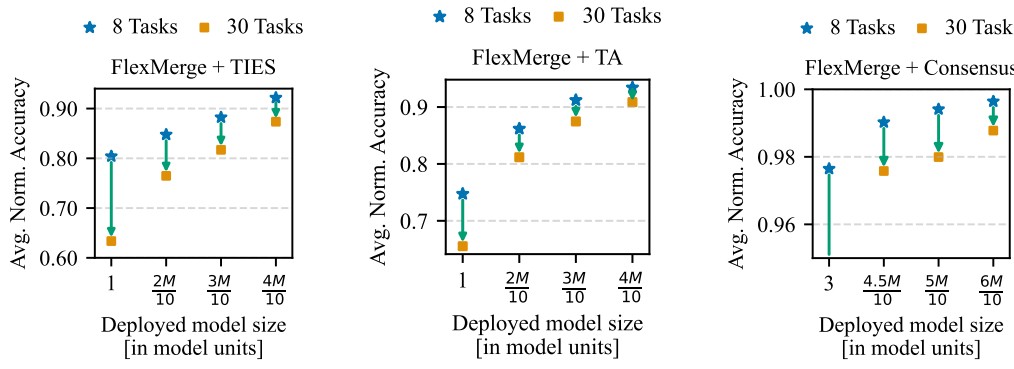

Figure 18: Deploying model sizes in proportion to the number of tasks ($M$) scales better than fixed size models ($1\times$ or $3\times$) as shown.

As the number of tasks increases, maintaining high accuracy in the merged model becomes challenging. While newer methods enhance accuracy scaling, they encounter limitations when the model size is fixed. Thanks to flexible model merging, this scaling can be significantly improved if *the deployed model size is chosen in proportion to the number of tasks*. To illustrate this, we chart in Figure 18, the drop in average normalized accuracy from 8 tasks to 30 tasks, when deploying a model of fixed size $1\times$ vs sizes chosen proportionally to $M$. Noticeably, the drop becomes smaller as the proportions increase, becoming $< 5\%$ at size $4M/10$ from $> 16.5\%$ at size of $1\times$ for TIES-MERGING.

We observe similar results across other algorithms. For TA at size $1\times$, the average normalized accuracy drops by $9.2\%$ when scaling from 8 to 30 tasks. With FLEXMERGE, this degradation can be significantly mitigated by adjusting the deployed model size in proportion to the number of tasks ($M$). For instance, setting the deployed model size to $3M/10$ reduces the drop to $3.7\%$, and increasing it to $4M/10$ brings the drop down further to $2.5\%$. Even for advanced algorithms such as CONSENSUS, fixed-size models can suffer substantial degradation. At a fixed size of $3\times$, the accuracy drop from 8 to 30 tasks for CONSENSUS is $14.5\%$ (beyond the y-axis limits of the figure). However, increasing the deployed model size to $4.5M/10$ reduces the drop to $2.5\%$, and to less than $1\%$ at $6M/10$. While fixed-size models struggle to scale effectively with the number of tasks, our results highlight the need to rethink accuracy scaling—advocating for model sizes that grow proportionally with task count.

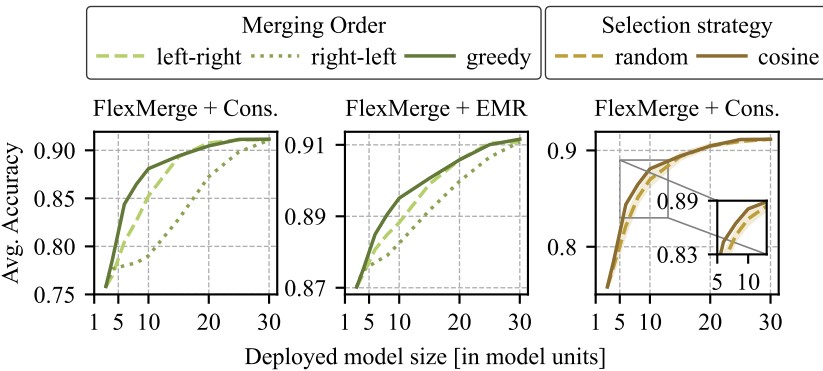

Figure 19: Greedy emerges as the best over left-to-right and right-to-left for both FLEXMERGE + CONSENSUS and FLEXMERGE + EMR-MERGING whereas cosine performs better than random.

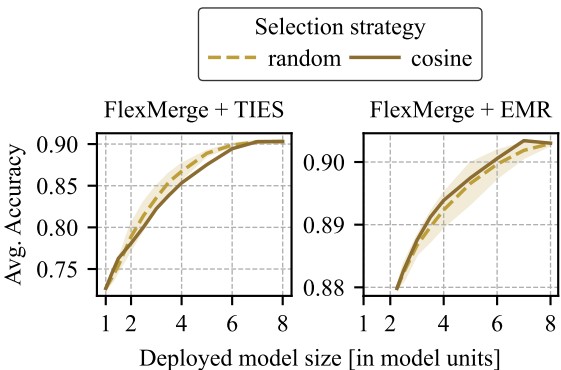

Figure 20: For TIES-MERGING, its trimming component makes similarity comparisons noisy. Hence, random selection performs slightly better than cosine. In contrast, cosine similarity performs better on EMR-MERGING and most other cases (see also Figures 6 and 19).

### C.8 ABLATIONS OF THE MERGING PROCEDURE

In this section, we include the remaining ablation results. In particular, we explore the impact of merging order on CONSENSUS and EMR-MERGING. We consider three merging orders: left-to-right, right-to-left and greedy. In the left-to-right order, we execute pairwise iterative merging using cosine similarity in block 1, followed by block 2, block 3 and so on (see Figure 1). Hence, all task-blocks in block $i$ are fully merged before carrying out any merging in block $i+1$. The right-to-left merging does exactly the same, but in reverse order, starting from the last block. In contrast to both, greedy merging can select any pairs in any block depending upon their cosine similarity, without any restriction on the order. Figure 19 shows the results for FLEXMERGE + CONSENSUS and FLEXMERGE + EMR-MERGING. Similar to our results in Figure 6, right-to-left performs the worst. As final layers tend to be more specialized, merging them first significantly hurts accuracy. Intuitively, left-to-right merging seems ideal. However, strictly following this order proves too rigid, and a more flexible greedy merging approach performs best. We further analyze the ordering in greedy in Section C.10.

We also explore the impact of random selection over cosine similarity based selection on more merging algorithms, namely TIES-MERGING and EMR-MERGING, extending our results in Figure 6. In Figure 19, cosine performs better than random on FLEXMERGE + CONSENSUS. However in Figure 20, random selection shows slightly better performance for FLEXMERGE + TIES-MERGING. This is likely due to the trim component of TIES-MERGING which makes similarity comparisons noisy. As detailed in Section B.1, we retain only the top $10\%$ parameters in the full task vectors before beginning the bottom-up merging process when using TIES-MERGING. While sparsification enables

better merging due to reduced interference Yadav et al. (2023), it also renders similarity comparisons noisy, leading to a slight drop in performance compared to random selection. With FLEXMERGE + EMR-MERGING, random shows high variance, and once again cosine performs better on average than random. In summary, cosine is generally superior across most cases in Figures 6, 19 and 20.

## C.9 RECONSTRUCTION LATENCY OF MASKING BASED APPROACHES

### C.9.1 HOW DO WE MEASURE THE RECONSTRUCTION LATENCY?

Approaches such as CONSENSUS and EMR-MERGING merge into a unified task vector $\boldsymbol{\tau}_{\text{uni}}$ and task-specific masks $(\boldsymbol{m}_1, \ldots, \boldsymbol{m}_M)$. To reconstruct the task-specific model, they also store $\boldsymbol{\theta}_{\text{pre}}$ and reconstruct as follows:

$$\hat{\boldsymbol{\theta}}_t = \boldsymbol{\theta}_{\text{pre}} + \boldsymbol{\tau}_{\text{uni}} \circ \boldsymbol{m}_t \tag{2}$$

Under FLEXMERGE, the size of the deployed model varies. Consider for a certain task $t$, two sets – $B_t^F$ and $B \backslash B_t^F$, comprising the set of all blocks that were fused and retained as original respectively. Notice the subscript $t$ which indicates that these sets could be different for different tasks depending on how the greedy fusion occurred. For each fused block $b \in B_t^F$, let $\mathcal{T}^b$ denote the subset of tasks which got fused. Then FLEXMERGE will store the corresponding $\boldsymbol{\tau}_{\text{uni}}^b$ along with the task specific masks $\{\boldsymbol{m}_k^b\}_{k \in \mathcal{T}^b}$ and $\boldsymbol{\theta}_{\text{pre}}^b$. Under FLEXMERGE, we then reconstruct as follows:

$$\text{for } b \in B_t^F : \hat{\boldsymbol{\theta}}_t^b = \boldsymbol{\theta}_{\text{pre}}^b + \boldsymbol{\tau}_{\text{uni}}^b \circ \boldsymbol{m}_t^b \tag{3}$$

And, no reconstruction is required for unfused blocks:

$$\text{for } b \in B \backslash B_t^F : \hat{\boldsymbol{\theta}}_t^b = \boldsymbol{\theta}_t^b \tag{4}$$

The reconstruction latency therefore depends upon the size of $B_t^F$. Ideally, we would measure the reconstruction latency as the time required to execute Equation (3) averaged across all tasks $t \in [M]$. However, this would mean that the reconstructed parameters $\hat{\boldsymbol{\theta}}_t$ occupy additional storage alongside $\boldsymbol{\theta}_{\text{pre}}$. To restrict this extra storage, we assume that the reconstruction happens in-place on $\boldsymbol{\theta}_{\text{pre}}$ as follows:

$$\text{for } b \in B_t^F : \boldsymbol{\theta}_{\text{pre}}^b = \boldsymbol{\theta}_{\text{pre}}^b + \boldsymbol{\tau}_{\text{uni}}^b \circ \boldsymbol{m}_t^b \tag{5}$$

Once the inference for task $t$ is completed, the pre-trained parameters are restored back to be ready for the next reconstruction:

$$\text{for } b \in B_t^F : \boldsymbol{\theta}_{\text{pre}}^b = \boldsymbol{\theta}_{\text{pre}}^b - \boldsymbol{\tau}_{\text{uni}}^b \circ \boldsymbol{m}_t^b \tag{6}$$

Thus, we report the reconstruction latency to be the total time required to execute both Equation (5) and Equation (6), averaged across all tasks.

### C.9.2 HOW DOES INCREASING THE DEPLOYED MODEL SIZE LOWER THE RECONSTRUCTION LATENCY?

As noted earlier, the reconstruction latency depends on the size of $B_t^F$. As the deployed model size progressively increases, more and more blocks move from $B_t^F$ to $B \backslash B_t^F$, and the time to reconstruct consequently reduces. The lowest deployed size *i.e.*, where all blocks are fused, $B_t^F = B$, incurs the highest reconstruction overhead. Conversely, the maximum possible deployed size (of $M \times$), with $B_t^F = \emptyset$, incurs zero reconstruction latency. Thus by employing larger deployed models generated by FLEXMERGE, practitioners can effectively reduce the reconstruction overhead in time-critical applications.

**Results.** Figure 21 shows the average reconstruction latency for CONSENSUS, measured on both CPU and GPU. The latency for EMR-MERGING closely matches that of CONSENSUS, as the additional scalar multiplications in EMR-MERGING incur negligible overhead; the dominant cost arises from the application of masks. As illustrated in Figure 21, merging into larger models using FLEXMERGE can significantly reduce reconstruction latency. For ViT-L/14 on CPU, latency drops from 240 ms to 0 ms, and for ViT-B/32, from 60 ms to 0 ms, depending on the deployed model size. On GPU, the latency similarly decreases—from 6 ms to 0 ms for ViT-L/14 and from 4 ms to 0 ms for ViT-B/32. To put this in perspective, the forward pass takes 12 ms for ViT-L/14 and 7 ms for ViT-B/32. On GPU, reconstruction latency can exceed 50% of the forward pass time, while on CPU, the overhead is several times higher. These reconstruction costs can be significant in time-critical scenarios. By enabling fine-grained control over the deployed model size, FLEXMERGE allows CONSENSUS and EMR-MERGING to achieve not only higher accuracy but also lower reconstruction latency.

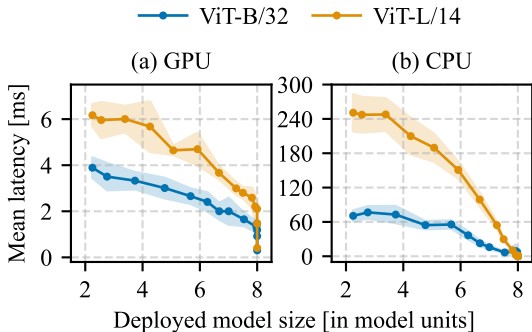

Figure 21: FLEXMERGE enables lowering of the reconstruction latency overheads for approaches such as CONSENSUS Wang et al. (2024) and EMR-MERGING Huang et al. (2024) by offering flexible control over the deployed model size. The default overhead at the lowest size can be significant, exceeding 50% for the forward pass time, which is $12\,\mathrm{ms}$ for ViT-L/14 and $7\,\mathrm{ms}$ for ViT-B/32 on GPU.

## C.10 MERGING ORDER ANALYSIS

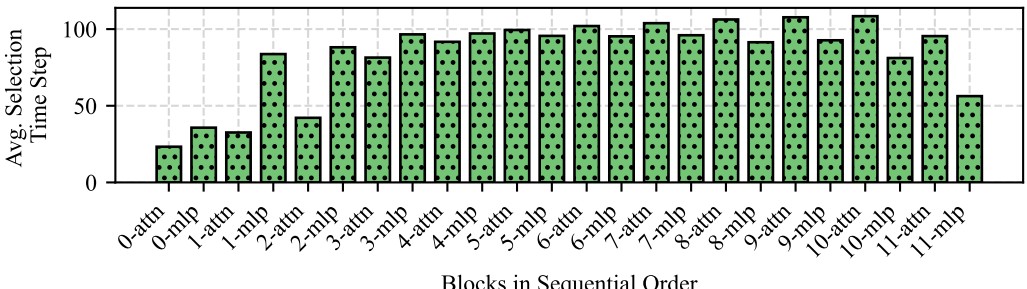

Figure 22: Visualizing the order of greedy selection blocks for FLEXMERGE + TA on 8 task vision benchmark with ViT-B/32. We focus only on Attention and MLP blocks as they share the biggest portion of the size of the model. We observe that the initial and final blocks get selected ahead of the intermediate blocks on average, indicated by the smaller values of their average selection time step.

Figure 22 visualizes the order of greedy block selection for the run corresponding to FLEXMERGE + TA on 8 task vision benchmark with ViT-B/32 (shown in Figure 3). We focus only on Attention and MLP layers as they share the biggest portion of the size of the model. There are 12 transformer layers in the ViT-B/32 model, resulting in a total of 24 blocks that we consider in the sequential order. The number of possible merges total to $(M-1) \times 24 = 7 \times 24 = 168$, with at most $(M-1)$ merges possible for each of the 24 blocks. The time steps are assigned in order of selection starting from 0 to 167. We report the average selection time step for every block in Figure 22. We observe that the initial and final blocks are, on average, selected earlier than the intermediate blocks, as indicated by their smaller average selection time step. While the overall trend aligns with the common understanding that earlier layers in neural networks learn general features shared across tasks and later layers capture more task-specific features with lower correlation, we find that the final few blocks may also exhibit higher cosine similarity and be selected early for merging. This flexibility, enabled by FLEXMERGE's greedy selection process, facilitates the efficient trade-off between model size and accuracy.

## C.11 MAIN RESULTS PRESENTED WITH NORMALIZED ACCURACY

We presented results using average accuracy in Figures 3 and 4. For completeness, Figures 23 and 24 show the corresponding results using average normalized accuracy, computed by dividing the

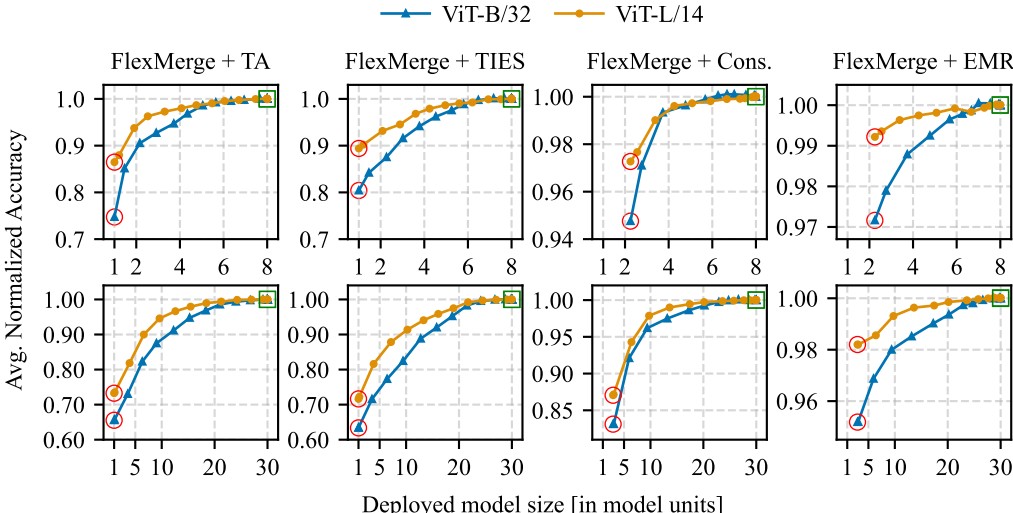

Figure 23: Vision benchmark results (Figure 3) shown with average normalized accuracy.

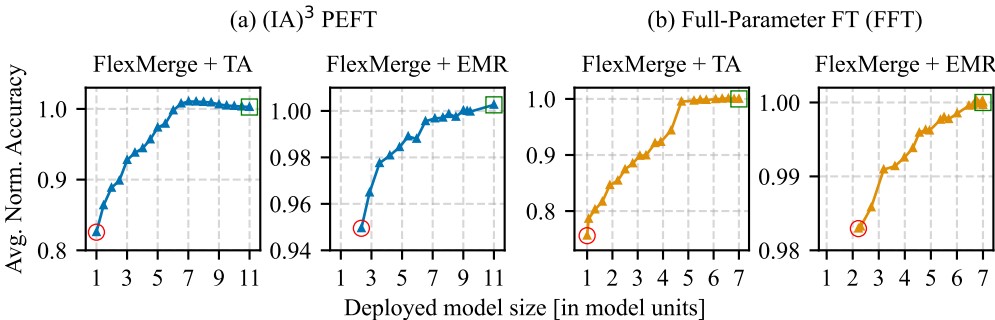

Figure 24: PEFT and FFT benchmark results (Figure 4) shown with average normalized accuracy.

accuracy of the merged model by the fine-tuning accuracy on each task, and then averaged across tasks.

### C.12 DATASET-WISE RESULTS

Figures 25 to 27 chart the dataset-wise results for FLEXMERGE + TA, corresponding to the plots in Figures 3 and 4.

## D COMPUTE RESOURCES

All experiments were conducted on an internal compute cluster comprising 2 x AMD EPYC 7543 32-Core 2.8GHz CPU Processor, equipped with 8 x NVIDIA A100 SXM4 80GB GPU. All of our experiments individually use only 1 out of the 8 GPU units. While the merging itself is efficient, the evaluation of test accuracy consumes bulk of the time and compute. The wall-clock time can range anywhere between $3\,\mathrm{h}$ for $8$ tasks to up to $12\,\mathrm{h}$ for $30$ tasks under the ViT-L/14 model for evaluating up to 20 merged sizes in the size range $[1, 30]$. Similarly, the wall clock times range up to $2\,\mathrm{h}$ for full-parameter fine-tuning, up to $8\,\mathrm{h}$ for PEFT and up to $5\,\mathrm{h}$ for multi-modal test evaluations. Across all experiments presented in this article, we estimate the total virtual CPU and GPU time to be approximately $1200\,\mathrm{h}$ each.

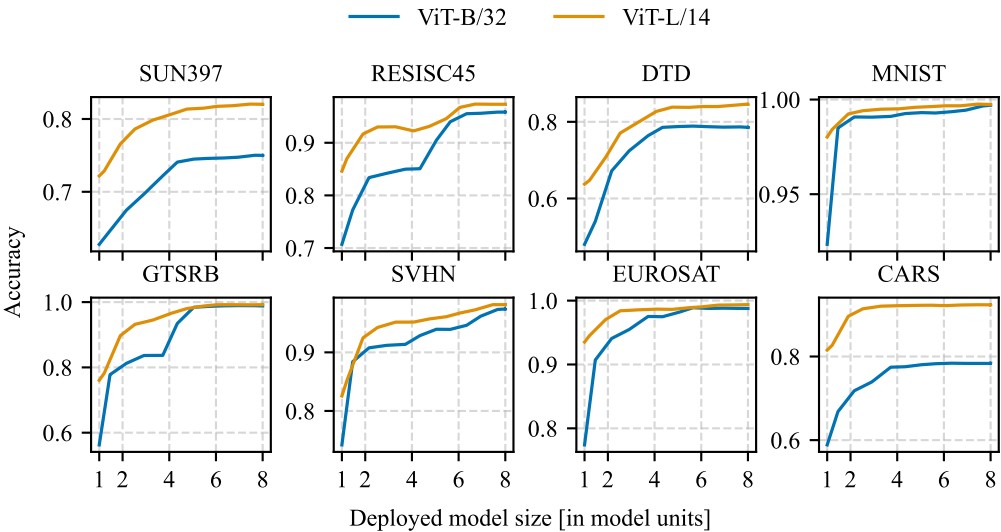

Figure 25: Dataset-wise results for FLEXMERGE + TA on the vision benchmark (Figure 3).

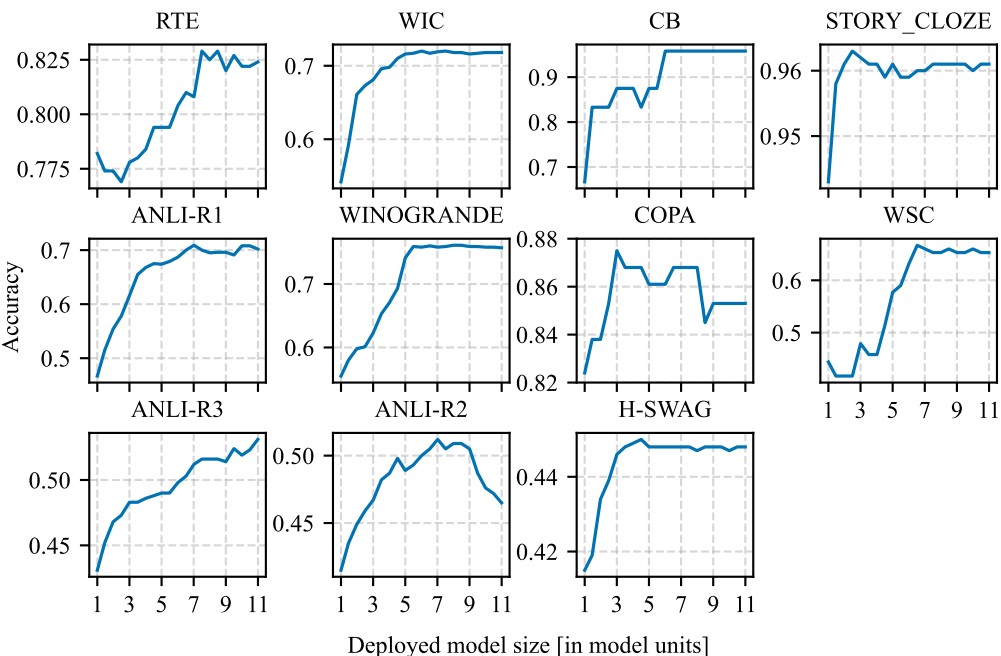

Figure 26: Dataset-wise results for FLEXMERGE + TA on the PEFT benchmark (Figure 4(a)).

# E   LLM USAGE STATEMENT

We acknowledge the use of LLMs in this work, limited to coding assistance, identifying potentially relevant related work, and improving the clarity and grammar of the manuscript. All LLM-generated content was reviewed and verified by the authors.

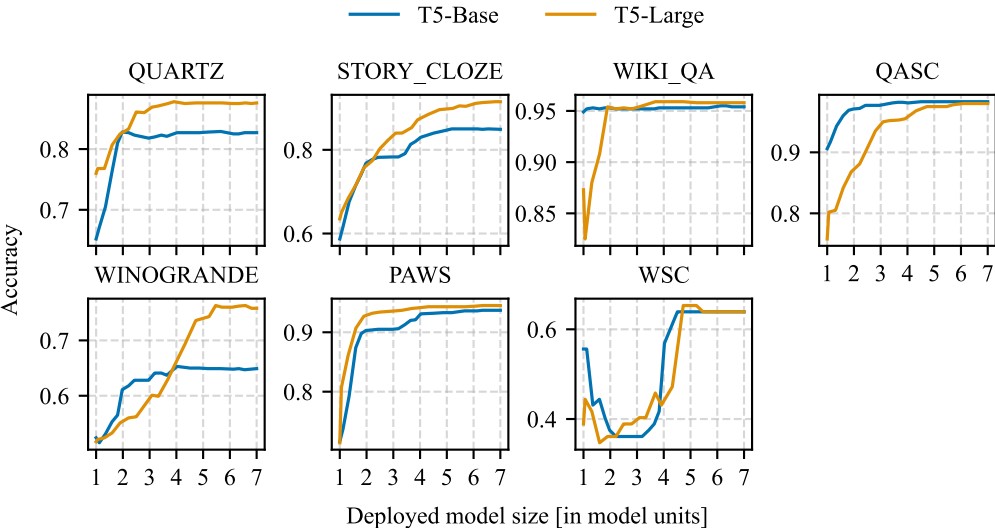

Figure 27: Dataset-wise results for FLEXMERGE + TA on the FFT benchmark (Figure 4(b)).

