# OpenReview forum: "Navigating the Accuracy-Size Trade-Off with Flexible Model Merging"
_ICLR.cc/2026/Conference — ICLR 2026 Poster_

### Official Review · Reviewer_jaXQ · 2025-10-28

**Soundness:** 2
**Presentation:** 2
**Contribution:** 2
**Rating:** 6
**Confidence:** 4

**Summary:**

This paper proposes a data-free model fusion framework called FlexMerge, aimed at addressing the trade-off between accuracy and model size in multi-task model fusion. FlexMerge supports generating fusion models of any size (including non-integer multiples) and can integrate various data-free fusion algorithms within a unified framework. Through experiments, the authors pointed out: moderately increasing the model size can bring significant accuracy improvements, and the performance rankings of different fusion algorithms are inconsistent at different sizes.

**Strengths:**

S1: The author systematically studied the global trade-off between accuracy and size in model fusion for the first time, breaking through the limitations of traditional 'single-model fusion'.
S2: The paper proposes a data-free framework supporting the fusion of model with non-integer sizes.
S3: The author reveals that the phenomenon of "algorithm performance re-ranking under different model sizes.

**Weaknesses:**

W1: Lack of insight. Although the experiments are thorough, there is a lack of theoretical explanation and further analysis on 'why certain algorithms perform better at larger scales'.
W2: Lack of in-depth discussion on certain algorithms. Methods such as Consensus and EMR require storing masks or reconstructing them during inference. Although FlexMerge can mitigate this, its additional overhead is not discussed in depth in the paper.
W3:Experiment: There is a lack of comparison with similar data-free methods on the same task.
W4: Lack of theoretical analysis. The algorithm uses a greedy strategy to merge blocks, and no explanation of optimality is given.
W5: Limited external validity. While the experimental suite is thorough within classification and homogeneous Transformer backbones, the evidence may not fully generalize to other settings. It would be helpful to include evaluations on detection/segmentation/generation tasks and on heterogeneous architectures to more convincingly demonstrate robustness across modalities and model families.

**Questions:**

Please refer to Weaknesses.

**Details Of Ethics Concerns:**

N.A.

---

> ### Author Response · Authors · 2025-11-20
>
> We sincerely thank the reviewer for their positive assessment of our work and constructive comments. We address each point in detail below.
>
> ----
>
> > W1] Lack of insight. Although the experiments are thorough, there is a lack of theoretical explanation and further analysis on 'why certain algorithms perform better at larger scales'.
>
> We thank the reviewer for appreciating the thoroughness of our experiments. We explain below why performance reranking happens at larger scales. As the scale is increased, we observe that simple algorithms like TA or Avg begin rivaling their advanced counterparts, such as TIES and PCB. This is because parameter interference between tasks greatly reduces as scale increases and parameters are disentangled. As a result, the effect of explicitly reducing parameter interference through advanced procedures such as TRIM in TIES or competition balancing in PCB offers little added benefit over reduced interference due to increase in scale itself. Thus, all algorithms remain within close margin of each other at larger scales despite starting from a large gap at smaller scales. We have included this explanation in "Cross-Algorithm Analysis" of Section 4.1 in our updated paper pdf.
>
> ----
>
> > W2] Lack of in-depth discussion on certain algorithms. Methods such as Consensus and EMR require storing masks or reconstructing them during inference. Although FlexMerge can mitigate this, its additional overhead is not discussed in depth in the paper.
>
> At inference time, FlexMerge does not introduce or incur additional overheads beyond those incurred by the respective merging algorithms of Consensus and EMR themselves, corresponding to reconstruction. Hence, we are unsure what the reviewer means by additional overhead. We have provided an in-depth discussion and analysis of how FlexMerge mitigates the reconstruction latency of Consensus and EMR in Appendix C.9. We are happy to clarify if the reviewer could elaborate the question further.
>
> ----
>
> > W3] Experiment: There is a lack of comparison with similar data-free methods on the same task.
>
>
> We have performed extensive assessments and reported cross-algorithm analysis on the same task across several benchmarks. We request the reviewer for concrete suggestions on what is missing to better understand their comment.
>
> ----
>
> > W4] Lack of theoretical analysis. The algorithm uses a greedy strategy to merge blocks, and no explanation of optimality is given.
>
> Finding the optimal merged model for a given size is a challenging problem. We acknowledged this limitation in Section 5. More broadly, theoretical insights behind effective model merging are limited [1], which is also widely acknowledged in this field (see limitations in [2]). In this context, we are the first to systematically show the practical benefits of merging beyond the one-model regime, via our greedy merging framework. We believe our work provides a strong foundation for future work to explore optimality.
>
> ---
>
> > W5] Limited external validity. While the experimental suite is thorough within classification and homogeneous Transformer backbones, the evidence may not fully generalize to other settings. It would be helpful to include evaluations on detection/segmentation/generation tasks and on heterogeneous architectures to more convincingly demonstrate
> robustness across modalities and model families.
>
> Our work focuses on the Task Arithmetic family [3] of approaches for multi-task model merging which is designed for homogeneous architectures. Merging of heterogeneous architectures, although interesting, is considerably more challenging and requires different set of techniques altogether [4, 5]. This is currently out-of-scope, similar to existing work in this area.
>
> Lastly, we studied not only classification but also generation, under both parameter efficient fine-tuning (PEFT) and full-parameter fine-tuning (FFT), using several different model families. Essentially, question answering tasks, making the majority of our NLP evaluation, evaluate which of the possible answers is most likely to be generated by the model. We further conducted assessments on out-of-distribution (OOD) and multimodal settings. Altogether, we have performed extensive evaluations to demonstrate generality, following the widely accepted benchmarking practices in this field.
>
> We thus request the reviewer to revisit their assessment of this weakness in the light of above clarifications.
>
>
> # References
>
> [1] Ortiz-Jimenez, Guillermo et al. "Task arithmetic in the tangent
> space: Improved editing of pre-trained models." In NeurIPS 2023.
>
> [2] Lu, Zhenyi, et al. "Twin-merging: Dynamic integration of modular expertise in model merging." In NeurIPS 2024.
>
> [3] Ilharco, Gabriel, et al. "Editing models with task arithmetic." In ICLR 2023.
>
> [4] Imfeld, Moritz, et al. "Transformer fusion with optimal transport." In ICLR 2024.
>
> [5] Singh, Sidak Pal and Jaggi, Martin. "Model fusion via optimal transport." In NeurIPS 2020.

---

### Official Review · Reviewer_R12B · 2025-10-29

**Soundness:** 4
**Presentation:** 4
**Contribution:** 3
**Rating:** 6
**Confidence:** 4

**Summary:**

This paper proposes FlexMerge, a data-free model merging framework enabling flexible merged models across sizes (including non-integers) via greedy block merging. It balances accuracy-storage, unifies merging algorithms. Experiments on vision/NLP (up to 30 tasks) show steep accuracy gains with modest size increases, shifting algorithm rankings, and efficiency.

**Strengths:**

1.The paper is well organized and well written.

2.The authors present a well-motivated approach .

3.It conducts numerous experiments, validates the experimental results on models of various series and sizes, and covers a wide range of evaluation tasks.

**Weaknesses:**

1. **The necessity of the proposed architecture is not clearly justified.**
   According to previous findings from *DARE*[1], *Twin-Merge*[2], and *Consensus*[3], only a small portion of task-specific parameters truly contribute to performance, and task information can be effectively compressed through pruning techniques (e.g., SVD compression or masking).
   Given that the labels of the assumed dataset are known, the authors should compare *FlexMerge* with other methods under the same level of task information compression to demonstrate the superiority of their architecture.
   For example, the authors could compress the task vectors in *TA* using SVD and dynamically select vectors based on task labels for a fair comparison.

2. **In Figure 4(a), the performance of FlexMerge + TA decreases between 5 and 11 tasks.**
   What causes this degradation? The paper should provide an explanation or analysis of this phenomenon.

3. **In the OOD setting, task labels are not available.**
   How does *FlexMerge* decide which branch to activate in this case? The paper should clarify the mechanism used for branch selection under the OOD scenario.

[1]  Language models are super mario: Absorbing abilities from homologous models as a free lunch.

[2] Twin-merging: Dynamic integration of modular expertise in model merging.

[3] Localizing task information for improved model merging and compression.

**Questions:**

see weaknesses

---

> ### Author Response · Authors · 2025-11-20
>
> We sincerely thank the reviewer for the highly detailed and technically rigorous comments. We have carefully addressed each issue and provide comprehensive responses below.
>
> ----
>
> > W1] The necessity of the proposed architecture is not clearly justified. According to previous findings from DARE[1], Twin-Merge[2], and Consensus[3], only a small portion of task-specific parameters truly contribute to performance, and task information can be effectively compressed through pruning techniques (e.g., SVD compression or masking). Given that the labels of the assumed dataset are known, the authors should compare FlexMerge with other methods under the same level of task information compression to demonstrate the superiority of their architecture. For example, the authors could compress the task vectors in TA using SVD and dynamically select vectors based on task labels for a fair comparison.
>
> This is an extremely thoughtful remark. To understand better why our proposed architecture is needed, we explain below why the reviewer's proposed idea does not suffice.
>
> Let $(\hat{\tau}\_1, \hat{\tau}\_2, \ldots, \hat{\tau}\_M)$ denote the SVD compressed task vectors for all $M$ tasks. Indeed, these task vectors can contain only a small portion of parameters after heavy compression, while still maintaining good performance. However, the actual model to be used for a specific task $k$ still needs to be reconstructed from the pre-trained parameter $\theta_\textrm{pre}$ before every inference:
>
> $\hat{\theta}\_k = \theta_\textrm{pre} + \hat{\tau}\_k$
>
> This incurs non-negligible reconstruction overhead as the final task-specific model is not readily available in memory during inference. We measured this reconstruction overhead for Consensus in Figure 21 (Appendix C.9) and showed that it could be as large as 50% of the forward pass time. Thus, while one can simply store $(\theta_\textrm{pre}, \{\hat{\tau}\_k\}\_{k=1}^M)$, one has to pay the cost of reconstruction for every batch of inference. On the other hand, storing reconstructed $(\hat{\theta}\_1, \hat{\theta}\_2, \ldots, \hat{\theta}\_M)$ is infeasible since they are not sparse due to $\theta_\textrm{pre}$, which is not sparse. This motivates the design of FlexMerge which does not incur such overheads when combined with TA, TIES, etc. while demonstrating strong accuracy.
>
> Finally, FlexMerge can also be combined with compression based methods such as Consensus wherein it allows for the possibility of reconstructing some blocks while directly using the fine-tuned ones for others, to achieve significantly improved accuracy. For instance, continuing the same example above, for a specific task $k$, we might have
>
> $\hat{\theta}^b_k = \theta_\textrm{pre}^b + \hat{\tau}_k^b \quad \text{ for } b \in B^F_k$
>
> $\hat{\theta}^b_k = \theta_k^b \quad \text{ for } b \in B \backslash B^F_k$
>
> where $B^F_k$ refers to the set of blocks that undergo reconstruction using compressed task vectors while the rest $B \backslash B^F_k$ simply use the fine-tuned parameters of task $k$. This significantly improves the accuracy of Consensus as demonstrated in Figure 3.
>
> ----
>
> > W2] In Figure 4(a), the performance of FlexMerge + TA decreases between 5 and 11 tasks. What causes this degradation? The paper should provide an explanation or analysis of this phenomenon.
>
> That's a good observation. This is not a performance degradation but rather the power of FlexMerge to occasionally find merged models that even exceed the fine-tuned models' accuracies. The accuracy at size $11\times$ is that of the fine-tuned models, which we consider optimal (green square). The figure shows that the merged model around size  $7\times$ slightly exceeded this accuracy. This can happen due to cross-task knowledge sharing, where parameters of certain tasks mutually benefit each other when merged in a certain way. While this phenomemon is certainly desirable, it is also hard to find. As size increases beyond $7\times$, the benefit gradually disappears as the merged model disentangales into $11$ separate fine-tuned models, reaching the corresponding fine-tuning accuracy.

---

> ### Author Response · Authors · 2025-11-20
>
> > W3] In the OOD setting, task labels are not available. How does FlexMerge decide which branch to activate in this case? The paper should clarify the mechanism used for branch selection under the OOD scenario.
>
> Several popular methods in the literature operate under the assumption of task labels, such as EMR-Merging [1], Consensus [2] and even data-based methods such as Surgery [3]. All these methods share a common limitation -- inability to handle OOD data out-of-the-box due to the problem of branch selection. Thus, they do not report OOD performance. This is indeed a limitation of FlexMerge too.
>
> Nevertheless, to overcome this issue, we proposed to ensemble different branches of FlexMerge and showed that this strategy can achieve strong OOD performance. More precisely, let $\hat{f}\_k(\theta_\textrm{uni}; x)$ denote the task-specific model for task $k$ obtained from FlexMerge's merged parameters $\theta_\textrm{uni}$ with $x$ as the input. We obtain the corresponding output for OOD data by ensembling: $\hat{f}(\theta_\textrm{uni}; x) = \sum_{k \in [M]} \hat{f}\_k(\theta_\textrm{uni}; x)$. Our results show that the FlexMerge's merged model performs better than any single in-domain expert. Furthermore, as size increases, even the OOD performance improves. We included these results in Appendix C.6. Another approach to handle OOD data is to apply model routing [4, 5, 6] that can be used to infer the best-suited task ID for OOD tasks. Routers are trained using low-dimensional embedding vectors representing an image or a query, clustering them according to previously seen embeddings and their corresponding task IDs. Applying these routing methods is beyond the scope of the present work, and would be an interesting future direction.
>
> # References
>
> [1] Huang, Chenyu, et al. "Emr-merging: Tuning-free high-performance model merging." In NeurIPS 2024.
>
> [2] Wang, Ke, et al. "Localizing Task Information for Improved Model Merging and Compression." In ICML 2024.
>
> [3] Yang, Enneng, et al. "Representation Surgery for Multi-Task Model Merging." In ICML 2024.
>
> [4] Zhuang, Richard, et al. "EmbedLLM: Learning compact representations of large language models." In ICLR 2025.
>
> [5] Ding, Dujian, et al. "Hybrid llm: Cost-efficient and quality-aware query routing." In ICLR 2024.
>
> [6] Hu, Qitian Jason, et al. "Routerbench: A benchmark for multi-llm routing system." ArXiv preprint 2024.

---

### Official Review · Reviewer_EyDU · 2025-10-31

**Soundness:** 2
**Presentation:** 2
**Contribution:** 1
**Rating:** 2
**Confidence:** 4

**Summary:**

The authors propose FlexMerge, a unified framework for data-free, block-level greedy model merging, which enables the combination of multi-task fine-tuned models at arbitrary scales from 1× to M×. The paper systematically demonstrates a favorable trade-off between accuracy and model size across vision, NLP, and multimodal scenarios, showing that scaling from 1× to 2× typically yields substantial accuracy improvements, while the performance approaches the single-task fine-tuning upper bound at scales far below M×, a phenomenon consistently observed across multiple merging algorithms.

**Strengths:**

1.The paper reframes model merging from a fixed single-target (1×) setup into a controllable continuous spectrum of model scales. This data-free formulation reflects real-world trade-offs between accuracy and resource budgets, turning a theoretical question into a deployable engineering solution.

2.A block-wise greedy strategy decomposes Transformers into Attention, MLP, and other units, enabling fine-grained control over model capacity. The proposed pluggable meta-framework allows the merging primitive  to be instantiated with different data-free algorithms (e.g., TA, TIES, Consensus), demonstrating strong versatility and generality.

3.Experimental results across ViT-B/32 and ViT-L/14 (covering 8 and 30 task settings), as well as T5/T0-3B under both PEFT and FFT configurations and multimodal scenarios, show consistent trends: performance improves sharply from 1× to 2× capacity and then saturates, confirming the framework’s stability and scalability across merging algorithms.

**Weaknesses:**

1.Although the proposed block-wise greedy merging strategy demonstrates strong empirical performance, the paper offers no theoretical justification or formal analysis (e.g., approximation guarantees or error bounds). Consequently, it remains unclear under what conditions this greedy selection strategy converges to an effective merging configuration, or what its worst-case behavior might be under large inter-task divergence.

2.All experiments were conducted on models sharing identical architectures. The method’s feasibility and performance under heterogeneous or partially aligned architectures (e.g., differing scales or network structures) remain unexplored, despite such configurations being common in real-world multi-source settings. This assumption limits the method’s generalizability and real-world applicability.

3.Although the paper reports partial results on merging time, inference latency, and reconstruction delay, the overall analysis remains insufficient. There is a lack of explicit complexity characterization for the block-level similarity computation and greedy matching procedure, as well as no quantitative comparison of memory and storage costs across different model scales and merging primitives.

**Questions:**

1, How about the experiments on merging LLMs ?

2, Definition & Construction of Fractional Models：

The paper introduces fractional model units (e.g., 2.25→), but the mechanism remains unclear.

How are fractional models constructed at the parameter level?

Does “fractional” refer to partial layers, partial parameter blocks, or continuous interpolation over task vectors?

How do you ensure structural consistency and avoid breaking model geometry when only a “fraction” is used?

A more precise definition is needed to understand how fractional capacity is achieved.

3: Merging multiple fine-tuned models is known to cause representational interference. This risk may increase with fractional merging.

4: How do you ensure stability and avoid representation collapse when using fractional model portions?

5: Do fractional models exhibit smooth performance scaling or sudden drops at certain fractional ratios?

---

> ### Author Response · Authors · 2025-11-20
>
> We sincerely thank the reviewer for their feedback and thoughtful comments. We noticed that several questions raised by the reviewer primarily stem from a confusion regarding the definition and construction of fractional models. Because our approach never operates on partial parameters, but only on full blocks, many of the issues raised (e.g., structural consistency, collapse, interference) do not arise in practice. We therefore provide a detailed clarification in Q2, before addressing questions Q3-Q5. We begin with our clarifications on the weakness raised.
>
> ----
>
> > W1] Although the proposed block-wise greedy merging strategy demonstrates strong empirical performance, the paper offers no theoretical justification or formal analysis (e.g., approximation guarantees or error bounds). Consequently, it remains unclear under what conditions this greedy selection strategy converges to an effective merging configuration, or what its worst-case behavior might be under large inter-task divergence.
>
> To the best of our knowledge, no existing work cited in our paper provides formal guarantees on worst case merging performance. More broadly, theoretical insights into when model merging should succeed or fail remain limited, a challenge widely acknowledged in the literature (see limitations in [1]). However, the absence of formal bounds has not impeded foundational empirical research in this field, with several methods being developed that improve the state-of-the-art, following the introduction of Task Arithemtic [2]. In this context, we believe our work makes a valuable contribution by providing the first systematic empirical characterization of the accuracy–size trade-off, revealing actionable insights into how data-free merging can overcome the performance limitations of the 1× regime.
>
> Moreover, the design of our method is well grounded in heuristic block selection based on cosine similarity, a popular metric in the literature [2, 3]. We have performed extensive ablations, demonstrated the versatility and generality of our approach across diverse setups, following widely accepted standard benchmarking practices in this field. We are grateful that this was also noted by the reviewer as one of the strengths of our paper.
>
> We therefore kindly request the reviewer to revisit their assessment of this weakness, given that *(i)* the absence of worst-case theoretical guarantees is a fundamental limitation shared by all existing methods, and *(ii)* our work advances the state of empirical understanding through new insights, extensive validation, and a practical framework that addresses core shortcomings of current data-free merging approaches.
>
> ----
>
> > W2] All experiments were conducted on models sharing identical architectures. The method’s feasibility and performance under heterogeneous or partially aligned architectures (e.g., differing scales or network structures) remain unexplored, despite such configurations being common in real-world multi-source settings. This assumption limits the method’s generalizability and real-world applicability.
>
> This is a well-known limitation of approaches based on Task Arithmetic [2], as we have rightfully noted in Section 5. Indeed, while merging heterogeneous models is interesting, it is generally considered more challenging and requires different set of approaches altogether [4, 5]. While this a nice future direction, it is currently out-of-scope of our study, as is also the case for existing work in this area.
>
> ----
>
> > W3] Although the paper reports partial results on merging time, inference latency, and reconstruction delay, the overall analysis remains insufficient. There is a lack of explicit complexity characterization for the block-level similarity computation and greedy matching procedure, as well as no quantitative comparison of memory and storage costs across different model scales and merging primitives.
>
> Thank you for the suggestion. The time complexity of FlexMerge is $O(M^2 * B * d_\text{max})$ where $M, B$ and $d_\text{max}$ refer to the number of tasks, number of blocks in the model and the maximum dimension of any block task vector respectively. We have updated the manuscript with a detailed complexity analysis deriving the aforementioned complexity, included in Section 3.2 and highlighted in blue.
>
> In practice, the merging time is quite small as we illustrate in Table 2 (Section 4.3). For our largest setup comprising $M = 30$ tasks on ViT-L/14 $(B = 24)$, the merging completes in merely 31 secs. Thanks to its training-free and data-free nature, our approach is significantly cheaper than contemporary data-based approaches that require solving expensive optimization objectives [12, 13, 14]. Additionally, this is a one-time cost -- it does not scale with the number of inference queries.

---

> ### Author Response · Authors · 2025-11-20
>
> > Q1] How about the experiments on merging LLMs ?
>
> We appreciate the reviewer's interest in LLM experiments. We would like to clarify that our evaluation already includes extensive experiments on language models across multiple scales, settings, and the most common fine-tuning setups studied in the model merging literature. We experiment with language models of up to 3 billion parameters (T0-3B) with parameter-efficient fine-tuning. These models are large enough to demonstrate meaningful capabilities while still being feasible for practitioners to deploy them in multi-task scenarios.
>
> Furthermore, our experimental setup closely follows widely accepted benchmarks in the model merging literature. We adopt the same models, tasks, and evaluation protocols from prior work [6, 9], covering vision, NLP and multi-modal setups. This standardized evaluation has been consistently used across recent model merging papers at top venues [10, 11]. Moreover, the thoroughness of our experimental evaluation was explicitly recognized as a strength by multiple reviewers.
>
> We therefore argue that our current experiments, following established practices in the field and covering multiple scales and settings, provide sufficient evidence for the effectiveness and generality of FlexMerge.
>
> ----
>
> > Q2] How are fractional models constructed at the parameter level? Does “fractional” refer to partial layers, partial parameter blocks, or continuous interpolation over task vectors? How do you ensure structural consistency and avoid breaking model geometry when only a “fraction” is used? A more precise definition is needed to understand how fractional capacity is achieved.
>
> Thank you for the question. We clarify the construction process below.
>
> We assume that each model is composed of $B$ sequential blocks where each block could be a layer or group of layers. Fractional models are constructed at block-level, and not directly at the parameter-level.
>
> **Illustrative Example.** We refer to Figure 1(a) to explain the fractional model size. For the simplicity of explanation, assume that all blocks have same size as shown in Figure 1(a). The fractional refers to, roughly speaking, number of blocks in the merged model over the number of blocks in one model. In Figure 1(a), each fine-tuned model has $B=4$ blocks and there are $M=5$ total tasks. Without any merging, there are $M \times B = 4 \times 5 = 20$ total blocks. Exisiting work always merges into a one model i.e. the 20 blocks are always merged into 4 blocks. However, FlexMerge permits merging into any number of desired blocks. For instance, after a few merging iterations of FlexMerge, there remain 7 total blocks as shown in the figure. The net size of this merged model as a factor of one model is therefore  $\frac{7 \times \textrm{block size}}{4 \times \textrm{block size}} = \frac{7}{4} = 1.75.$
>
> In practice, each block could have different size. We account for these size differences in our framework by refering to the size of each block $b$ as $\textrm{size}(\tau^b)$ (line 6, Algorithm 1) where $\tau^b$ is the task vector for the block. In this case, the size ratio is given as $\frac{\sum_{b \in \textrm{Merged Model}} \textrm{size}(\tau^{b})}{\sum_{b \in [B]} \textrm{size}(\tau^b)}$ where the denominator evaluates the total size of blocks in one model. This ratio $\in [1, M]$ where on one end, the merged model has exactly the same number of blocks as one model; and on the other end, it has all blocks of all models (i.e. no merging). Finally, structural consistency of the model is automatically maintained since we always operate at the granularity of well-defined blocks.
>
> We hope this helped clarify the reviewer's question. We are happy to elaborate if there are any further questions.

---

> ### Author Response · Authors · 2025-11-20
>
> > Q3] Merging multiple fine-tuned models is known to cause representational interference. This risk may increase with
> fractional merging.
>
> With reference to our clarification above, merging into one model causes representational interference. In fact, FlexMerge reduces this interference by permitting more blocks than in a single merged model. As the number of blocks are increased, the interference is gradually reduced, leading to higher accuracy.
>
> ----
>
> > Q4] How do you ensure stability and avoid representation collapse when using fractional model portions?
>
> With reference to our clarification above, as fractional models still comprise well-defined blocks, we do not observe any representational collapse.
>
> ----
>
> > Q5] Do fractional models exhibit smooth performance scaling or sudden drops at certain fractional ratios?
>
> We consistently find that average accuracy smoothly improves as size is gradually increased from $1$ to $M$ across numerous benchmarks. This strongly demonstrates the correctness and generality of our algorithmic framework.
>
> # References
>
> [1] Lu, Zhenyi, et al. "Twin-merging: Dynamic integration of modular expertise in model merging." In NeurIPS 2024.
>
> [2] Ilharco, Gabriel, et al. "Editing models with task arithmetic." In ICLR 2023.
>
> [3] Zhang, Mingyang, et al. "Channel merging: Preserving specialization for merged experts." In AAAI 2025.
>
> [4] Imfeld, Moritz, et al. "Transformer fusion with optimal transport." In ICLR 2024.
>
> [5] Singh, Sidak Pal and Jaggi, Martin. "Model fusion via optimal transport." In NeurIPS 2020.
>
> [6] Yadav, Prateek, et al. "Ties-merging: Resolving interference when merging models." In NeurIPS 2023.
>
> [7] Huang, Chenyu, et al. "Emr-merging: Tuning-free high-performance model merging." In NeurIPS 2024.
>
> [8] Du, Guodong et al. "Parameter Competition Balancing for Model Merging" In NeurIPS 2024.
>
> [9] Wang, Ke, et al. "Localizing Task Information for Improved Model Merging and Compression." In ICML 2024.
>
> [10] Gargiulo, Antonio Andrea, et al. "Task singular vectors: Reducing task interference in model merging." In CVPR 2025.
>
> [11] Sun, Wenju, et al. "CAT Merging: A Training-Free Approach for Resolving Conflicts in Model Merging." In ICML 2025.
>
> [12] Yang, Enneng, et al. "Representation Surgery for Multi-Task Model Merging." In ICML 2024.
>
> [13] Yang, Enneng, et al. "AdaMerging: Adaptive Model Merging for Multi-Task Learning." In ICLR 2024.
>
> [14] Xu, Jing, Jiazheng Li, and Jingzhao Zhang. "Scalable Model Merging with Progressive Layer-wise Distillation." In ICML 2025.

---

> > ### Comment · Reviewer_EyDU · 2025-11-28
> >
> > Thank you for the detailed and carefully written rebuttal. I appreciate the authors’ clarifications, particularly the explanation regarding the construction and definition of fractional models, the block-level design choices, the complexity analysis, and the justification of the empirical nature of merging research. These responses addressed the majority of my concerns and resolved several misunderstandings I originally had.
> >
> > The exposition on structural consistency and interference under fractional merging was especially helpful, and the additional explanation of the heuristic rationale and empirical validation strengthened the credibility of the proposed approach. I am also satisfied with the clarifications around assumptions on architectural homogeneity and the positioning of this work within the broader literature on task arithmetic.
> >
> > My primary remaining concern lies with the absence of large-scale LLM merging experiments. As model merging is increasingly applied in the LLM setting, I believe that evaluating the proposed framework in this domain would significantly strengthen the claims of generality. That said, I acknowledge that the authors followed common benchmarks used in prior works and that the evaluation includes models up to the multi-billion parameter scale.
> >
> > Overall, the rebuttal addressed most of my issues clearly and convincingly. Given the strength of the empirical characterization, the novelty of fractional merging, and the improved clarity provided in the responses, I am raising my score to 4.

---

### Official Review · Reviewer_hggn · 2025-10-31

**Soundness:** 3
**Presentation:** 2
**Contribution:** 3
**Rating:** 6
**Confidence:** 4

**Summary:**

The paper introduces a block-level merging enhancement technique: it separates each expert into block groups and greedily merges the most similar pair within the same block using a given merging algorithm. During inference, it requires knowledge of the task ID and the specific block in which the task’s parameters reside, then routes the input to that block. The authors conduct extensive experiments on models across image domains (e.g., CLIP ViT‑B/32, ViT‑L/14) and PEFT/NLP setups.

**Strengths:**

- S1: This paper proposes a conceptually novel block-wise merging technique, serving as the interpolation between the single merged model and “retain‑everything”. It can balance performance and memory cost.
- S2: FlexMerge can plug into existing algorithms (TA, TIES, EMR, Consensus). The authors propose recomputing pairwise similarities and using a DSU structure to speed up the process.
- S3: The paper tests multiple base models (ViT‑B/32, ViT‑L/14), 8‑task and 30‑task suites, and PEFT scenarios (T0‑3B tasks), and conducts extensive ablation studies.

**Weaknesses:**

- **W1 (Major):** FlexMerging requires specific task iD. It needs a task ID to route task-specific inputs to the corresponding blocks, which imposes at least two constraints:
  - It cannot be applied in scenarios where we do not know which task the input originates from. Requiring a known task ID is unfair when comparing against methods that make no such assumption—e.g., task arithmetic or TIES.
  - It cannot handle out-of-domain evaluation, which is very common in practice. The method assumes that an input from task A must be assigned to the merged block containing the expert parameters for task A. If an unknown task B is presented, the system cannot assign it appropriately.

- **W2 (Major):** The computational complexity for $M$ tasks × $B$ blocks (involving pairwise similarity matrix computation and greedy merging steps) is not fully analyzed. For instance, with very large $M$ (e.g., 100+ tasks) and large $B$ (e.g., DeepSeek with 61 layers or Qwen with 94 layers), the merging cost in terms of memory and time remains unknown.

- **W3 (Minor):** The target size appears more empirical and may count different stored artifacts (e.g., masks for Consensus, per-block scalars for EMR). Choosing a size given a memory or accuracy target is heuristic.

**Questions:**

See weakness

**Details Of Ethics Concerns:**

See Weakness

---

> ### Author Response · Authors · 2025-11-20
>
> We sincerely thank the reviewer for their constructive comments and positive assessment of our work. We are highly encouraged by the reviewer's thoughtful remarks and provide detailed responses to all the questions below.
>
> ----
>
> > W1] FlexMerging requires specific task ID. It needs a task ID to route task-specific inputs to the corresponding blocks, which imposes at least two constraints:
>     - It cannot be applied in scenarios where we do not know which task the input originates from. Requiring a known task ID is unfair when comparing against methods that make no such assumption—e.g., task arithmetic or TIES.
>     - It cannot handle out-of-domain evaluation, which is very common in practice. The method assumes that an input from task A must be assigned to the merged block containing the expert parameters for task A. If an unknown task B is presented, the system cannot assign it appropriately.
>
> The two questions are related, and are handled (or not handled) depending on the family of the algorithm as we explain below.
>
> **Knowledge of Task-ID during inference:** For vision tasks, a task-ID is always required for all methods to correctly apply the classification-head belonging to the task [1, 2]. Moreover, beyond vision tasks, several popular methods in the literature operate under the assumption of task-ID, such as EMR-Merging [3] and Consensus [4]. They apply task-specific masks to reconstruct a task-specific models at the inference time. Even some data-based methods require task-ID assumption, such as Surgery [5] which trains task-specific adapters. Our method belongs to this family of methods, using a task-ID assumption.
>
> **OOD evaluation:** Methods requiring a task-ID indeed do not directly handle OOD data. For instance, EMR-Merging [3] does not handle OOD data, despite its high-performance and popularity. Nevertheless, we show that FlexMerge can handle OOD data by ensembling branches corresponding to different tasks. More precisely, let $\hat{f}\_k(\theta_\textrm{uni}; x)$ denote the task-specific model for task $k$ obtained from FlexMerge's merged parameters $\theta_\textrm{uni}$ with $x$ as the input. We obtain the corresponding output for OOD data by ensembling: $\hat{f}(\theta_\textrm{uni}; x) = \sum_{k \in [M]} \hat{f}\_k(\theta_\textrm{uni}; x)$. Our results show that the FlexMerge's merged model performs better than any single in-domain expert. Furthermore, as size increases, even the OOD performance improves. We included these results in Appendix C.6. Another approach to handle OOD data is to apply model routing [8, 9, 10] that can be used to infer the best-suited task ID for OOD tasks. Routers are trained using low-dimensional embedding vectors representing an image or a query, clustering them according to previously seen embeddings and their corresponding task IDs. Applying these routing methods is beyond the scope of the present work, and would be an interesting future direction.
>
> **Broader perspective:** The accuracy gap exhibited by the single merged model vs. fine-tuned models remains fairly high in data-free settings, despite the advances in the state-of-the-art. Thus improving in-domain performance in data-free settings has been our primary goal. A task-ID assumption facilitates this search for promising solutions. We acknowledge this limitation, yet we believe it offers a novel design space for pushing the state-of-the-art in data-free settings. We illustrated this through the design of FlexMerge, as also corroborated by existing work [3, 4]. Simulatenously, we also offer a promising way to handle OOD data, by ensembling different task branches in FlexMerge. While ensembling is not a perfect solution as it may incur higher cost, the strong positive results indicate that this could be a viable direction for future research. Another way to handle OOD data is through model routing methods that use clustering of embeddings corresponding to previous inference queries whose task IDs are known.

---

> ### Author Response · Authors · 2025-11-20
>
> > W2] The computational complexity for tasks × blocks (involving pairwise similarity matrix computation and greedy merging steps) is not fully analyzed. For instance, with very large M (e.g., 100+ tasks) and large B (e.g., DeepSeek with 61 layers or Qwen with 94 layers), the merging cost in terms of memory and time remains unknown.
>
> Indeed, the merging cost scales with $M$ and $B$. We derive the time complexity of FlexMerge to be $O(M^2 * B * d_\text{max})$ where $M, B$ and $d_\text{max}$ refer to the number of tasks, number of blocks in the model and the maximum dimension of any block task vector respectively. We have updated the manuscript with a detailed analysis deriving the aforementioned complexity, included in Section 3.2 and highlighted in blue.
>
> In practice, the merging time is quite small as we illustrate in Table 2 (Section 4.3). For our largest setup comprising $M = 30$ tasks on ViT-L/14 $(B = 24)$, the merging completes in merely 31 secs. Thanks to its training-free and data-free nature, our approach is significantly cheaper than contemporary data-based approaches that require solving expensive optimization objectives [5, 6, 7]. For instance, an AdaMerge optimization may take several hours for a much smaller scale (e.g. 2+ hours for ViT-B/32 under $M=8$) [Table 2, 6].
>
>
> To the best of our knowledge, we are not aware of any published benchmark that considers 100+ tasks on DeepSeek or Qwen models. While this is certainly interesting, we argue that the cost efficiency of our method is well justified by its data-free and training-free nature, as well as current evaluations on up to 30 tasks. Moreover, this is a one-time cost and it does not scale with the number of inference queries handled by the merged model.
>
>
> ----
>
> > W3] The target size appears more empirical and may count different stored artifacts (e.g., masks for Consensus, per-block scalars for EMR). Choosing a size given a memory or accuracy target is heuristic.
>
> That is a good point. Ideally (i.e. without explicitly measuring the accuracy-size trade-off), we expect a practitioner to merge into the highest model size that is permissible in a given deployment scenario. Once this size is determined, the practitioner can leverage FlexMerge to generate the model of corresponding size (inclusive of all stored artifacts artifacts depending on the choice of merging algorithm). On the other hand, if the accuracy-size trade-off can be measured, the target size can be more effectively adjusted considering all memory vs. performance constraints. We will add a discussion on this in the updated paper. Thank you for the suggestion!
>
> ----
>
> Please consider increasing your score after reading the above explanation. Thank you again for your constructive feedback and suggestions!
>
> # References
>
> [1] Ilharco, Gabriel, et al. "Editing models with task arithmetic." In ICLR 2023.
>
> [2] Yadav, Prateek, et al. "Ties-merging: Resolving interference when merging models." In NeurIPS 2023.
>
> [3] Huang, Chenyu, et al. "Emr-merging: Tuning-free high-performance model merging." In NeurIPS 2024.
>
> [4] Wang, Ke, et al. "Localizing Task Information for Improved Model Merging and Compression." In ICML 2024.
>
> [5] Yang, Enneng, et al. "Representation Surgery for Multi-Task Model Merging." In ICML 2024.
>
> [6] Yang, Enneng, et al. "AdaMerging: Adaptive Model Merging for Multi-Task Learning." In ICLR 2024.
>
> [7] Xu, Jing, Jiazheng Li, and Jingzhao Zhang. "Scalable Model Merging with Progressive Layer-wise Distillation." In ICML 2025.
>
> [8] Zhuang, Richard, et al. "EmbedLLM: Learning compact representations of large language models." In ICLR 2025.
>
> [9] Ding, Dujian, et al. "Hybrid llm: Cost-efficient and quality-aware query routing." In ICLR 2024.
>
> [10] Hu, Qitian Jason, et al. "Routerbench: A benchmark for multi-llm routing system." ArXiv preprint 2024.

---

### Author Response · Authors · 2025-12-03

We thank the reviewers and the Area Chairs for their time and constructive feedback. We would like to summarize improvements made to the manuscript as a result of the reviews, as well as our interactions with the reviewers during the discussion period. All changes to the original manuscript are highlighted in blue.

**Key revisions to the manuscript:**

-   Added time complexity analysis of FlexMerge in Section 3.2, showing it takes $O(BM²d_\text{max})$.
-   Added empirical merging time for ViT-L/14 models in Table 2, showing efficient merging time across various base model sizes.
-   Added clarification about reduced parameter interference at larger model sizes in "Cross-algorithm analysis" of Section 4.1.

**Among the main concerns addressed in our responses:**

-   **Computational complexity**  (Reviewers hggn, EyDU): We provided detailed time complexity analysis, now included in Section 3.2.
-   **Fractional model construction**  (Reviewer EyDU): We clarified that fractional models are constructed at the block level (not parameter level).
-   **Task-ID requirement and OOD handling**  (Reviewers hggn, R12B): We explained that task-ID is standard for vision classification and masking-based methods (EMR, Consensus), and demonstrated how FlexMerge can handle OOD via branch ensembling in Appendix C.6.
-   **Algorithm ranking insights**  (Reviewers jaXQ, EyDU): We explained that at larger scales, parameter interference naturally reduces, diminishing the benefit of explicit interference-reduction mechanisms in more advanced methods, and making simpler methods rival more sophisticated counterparts at larger model sizes.

Reviewer EyDU acknowledged that we addressed their concerns regarding fractional model construction, complexity analysis, block-level design choices, and the empirical nature of merging research, and as a result updated their score from 2 to 4 before the rollback of scores. We have not received responses from the remaining reviewers (hggn, R12B, jaXQ), all of whom rated the paper 6.

We believe our revisions have strengthened the manuscript and our responses have addressed the key technical concerns raised during the review process.

---

### Meta-Review · Area_Chair_7Mgu · 2026-01-06

**Summary:**

This paper proposes FlexMerge, a data-free and training-free model merging framework that enables generating merged models of varying sizes, spanning the full spectrum from a single merged model to retaining all individual fine-tuned models, realizing controllable and efficient trade-off between the model performance and model size. Experiments on vision, language, and PEFT models demonstrate the effectiveness of the proposed method.

Reviewers’ concerns mainly include:

1) the assumptions and scope of the proposed framework, particularly the reliance on task IDs at inference time and its implications for fairness of comparison and OOD applicability.
2) the scalability and efficiency of the block-level greedy merging procedure, as well as the clarity of the fractional model construction.
3) the lack of theoretical guarantees and initially limited analysis of deployment-time overhead compared to task-vector compression methods.
4) the absence of validation on mainstream large language models was highlighted as a remaining limitation for claims of generality.

**Reviewer Concerns:**

Overall, most of the concerns have been addressed during the rebuttal and discussion period, including:

(1) the complexity and efficiency analysis;
(2) the comparison between task-vector compression methods and deployment-time reconstruction overhead; and
(3) the clarification of the task-ID assumption and the proposed strategy for handling OOD inputs.

However, there are still some remaining or partially addressed concerns, including the lack of experiments on mainstream LLMs, which slightly weakens the claims regarding generality to modern LLM settings.

**Reviewer Scores:**

Initially, Reviewers hggn, R12B, and jaXQ each rated the paper 6, while Reviewer EyDU rated it 2.

After the rebuttal and discussion period, Reviewer EyDU indicated that his major concerns had been addressed and stated that he would raise their score to 4. The scores from other reviewers would remain around 6.

---

### Decision · Program_Chairs · 2026-01-26

Accept (Poster)